# Twenty years of change in benthic communities across the Belizean Barrier Reef

Catherine Alves[1,2], Abel Valdivia[3], Richard B. Aronson[4], Nadia Bood[5], Karl D. Castillo[6], Courtney Cox[3], Clare Fieseler[7], Zachary Locklear[8], Melanie McField[9], Laura Mudge[10,11], James Umbanhowar[1,10], John F. Bruno[10]*

**1** Environment, Ecology, and Energy Program, The University of North Carolina at Chapel Hill, Chapel Hill, North Carolina, United States of America, **2** ECS Federal, Inc., in support of Northeast Fisheries Science Center, Social Science Branch, National Oceanic and Atmospheric Administration, Narragansett, RI, United States of America, **3** Rare, Arlington, Virginia, United States of America, **4** Department of Ocean Engineering and Marine Sciences, Florida Institute of Technology, Melbourne, Florida, United States of America, **5** World Wildlife Fund Mesoamerica, Belize Field Programme Office, Belize City, Belize, Central America, **6** Department of Marine Sciences, The University of North Carolina at Chapel Hill, Chapel Hill, North Carolina, United States of America, **7** Science, Technology, and International Affairs Program, Georgetown University, Washington, District of Columbia, United States of America, **8** Green Bay Wildlife Conservation Office, United States Fish and Wildlife Service, New Franken, Wisconsin, United States of America, **9** Healthy Reefs for Healthy People Initiative, Smithsonian Institution, Fort Pierce, FL, United States of America, **10** Department of Biology, The University of North Carolina at Chapel Hill, Chapel Hill, North Carolina, United States of America, **11** Integral Consulting Inc., Annapolis, Maryland, United States of America

* jbruno@unc.edu

**Data Availability Statement:** All relevant code and data are available at here: https://github.com/calves06/Belizean_Barrier_Reef_Change.

**Funding:** This manuscript is based upon work supported by the National Science Foundation

## Abstract

Disease, storms, ocean warming, and pollution have caused the mass mortality of reef-building corals across the Caribbean over the last four decades. Subsequently, stony corals have been replaced by macroalgae, bacterial mats, and invertebrates including soft corals and sponges, causing changes to the functioning of Caribbean reef ecosystems. Here we describe changes in the absolute cover of benthic reef taxa, including corals, gorgonians, sponges, and algae, at 15 fore-reef sites (12–15m depth) across the Belizean Barrier Reef (BBR) from 1997 to 2016. We also tested whether Marine Protected Areas (MPAs), in which fishing was prohibited but likely still occurred, mitigated these changes. Additionally, we determined whether ocean-temperature anomalies (measured via satellite) or local human impacts (estimated using the Human Influence Index, HII) were related to changes in benthic community structure. We observed a reduction in the cover of reef-building corals, including the long-lived, massive corals *Orbicella* spp. (from 13 to 2%), and an increase in fleshy and corticated macroalgae across most sites. These and other changes to the benthic communities were unaffected by local protection. The covers of hard-coral taxa, including *Acropora* spp., *Montastraea cavernosa*, *Orbicella* spp., and *Porites* spp., were negatively related to the frequency of ocean-temperature anomalies. Only gorgonian cover was related, negatively, to our metric of the magnitude of local impacts (HII). Our results suggest that benthic communities along the BBR have experienced disturbances that are beyond the capacity of the current management structure to mitigate. We recommend that managers devote greater resources and capacity to enforcing and expanding existing marine

(DGE-1650116 to CA, OCE-0940019 to JFB, and partial support from OCE-1535007 to RBA), the Rufford Small Grant Foundation, the National Geographic Society, the International Society for Reef Studies/Center for Marine Conservation Reef Ecosystem Science Fellowship, the Elsie and William Knight, Jr. Fellowship from the Department of Marine Science at the University of South Florida, the J. William Fulbright program, the Organization of American States Fellowship, the World Wildlife Fund-Education for Nature Program, the Kuzimer-Lee-Nikitine Endowment Fund, the Nicholas School International Internship Fund at Duke University, the Lazar Foundation, and the Environment, Ecology and Energy Program, the Department of Biology, and the Chancellor's Science Scholar Research Fund at the University of North Carolina at Chapel Hill. Any opinions, findings, and conclusions or recommendations expressed in this material are those of the authors and do not necessarily reflect the views of the funders. The funders had no role in study design, data collection and analysis, decision to publish, or preparation of the manuscript. CA is currently employed at ECS Federal Inc., this agency played no role in this study. The funders provided support in the form of salaries for all authors, but did not have any additional role in the study design, data collection and analysis, decision to publish, or preparation of the manuscript. The specific roles of these authors are articulated in the 'author contributions' section.

**Competing interests:** The authors have declared that no competing interests exist. CA is currently employed at ECS Federal Inc. This does not alter our adherence to PLOS ONE policies on sharing data and materials

protected areas and to mitigating local stressors, and most importantly, that government, industry, and the public act immediately to reduce global carbon emissions.

## Introduction

Coral reefs worldwide have experienced remarkable changes over the past 40–50 years, particularly the widespread declines of reef-building corals and large, predatory fishes [1–7]. These changes have caused a reduction in or effective loss of essential ecological functions, including the provisioning of habitat for fisheries production and the maintenance of reef structure for shoreline protection [8, 9]. Given the substantial economic and cultural value of healthy reefs [10], this degradation is affecting coastal human communities that depend on reefs for food, income, and protection from storms.

Numerous factors are responsible for the well-documented degradation of Caribbean reefs. Acroporid corals, which dominated Caribbean reefs for thousands of years, experienced 90–95% mortality due to white-band disease in the 1980s [11]. This disease, likely exacerbated by ocean warming [12], coupled with increased frequency and intensity of hurricanes [13–15], reduced the habitat complexity, or rugosity, of Caribbean reefs [16]. Several other disease syndromes have greatly reduced the cover of other coral taxa, including black-band disease, which primarily affects brain corals [17]; yellow-band disease, which affects *Orbicella* spp. [18]; and, more recently, stony coral tissue loss disease, which affects numerous species, including *Dendrogyra cylindrus*, *Pseudodiploria strigosa*, *Meandrina meandrites*, *Eusmilia fastigiata*, *Siderastrea siderea* and *Diploria labyrinthiformis* [19]. Coral bleaching and other manifestations of ocean warming, including increased disease severity, are the primary causes of coral loss in the Caribbean [20–27]. On local scales, increased sedimentation and pollution from coastal development affect coral reefs by smothering corals and increasing turbidity [28, 29]. Secondary drivers of coral degradation include factors that have increased the cover of fleshy macroalgae (seaweeds), such as the death of scleractinian corals and the consequent opening of space and other resources [30], nutrient loading, and reduced herbivory. Herbivory has declined primarily because of the loss of the black sea urchin *Diadema antillarum* due a regional disease outbreak https://paperpile.com/c/IRBuuo/7Ngg+LZnc+XUQ0[31] and severe reductions of populations of herbivorous fishes due to fishing [32–37].

Despite the clear and well-documented changes to Caribbean reefs, there is ongoing disagreement about the causes of and best remedies for reef decline [20, 38–41]. The crux of the debate is about the relative importance of local causes—pollution, eutrophication, fishing, and consequent seaweed blooms—compared with regional-to-global causes such as ocean warming and acidification. Scientists, agencies, and organizations that view localized drivers as predominant generally argue for local mitigation, the primary recommendation being fisheries restrictions, such as within Marine Protected Areas (MPAs) [34, 42–44], and local reduction of pollution and other threats [45]. In contrast, the view that anthropogenic climate change has been a significant or the primary cause of reef decline, local impacts on resilience notwithstanding [46], leads to the conclusion that without rapid cuts in carbon emissions, local protections and other localized management actions, such as restoration, will ultimately fail [20, 39, 47].

The purpose of this study was to measure changes to benthic communities of the Belizean Barrier Reef (BBR) from 1997 to 2016 and determine whether they were related to protection status, local human impacts, and/or ocean-temperatures anomalies (i.e., ocean heatwaves). We

surveyed the coral reef benthos at 15 sites between 1997 and 2016 [48–50]. We found that benthic-community composition changed substantially during this period, and that the observed loss of corals was negatively related to ocean heatwaves but largely unaffected by local impacts or protection status.

## Materials and methods

### Study area and protection status

Scientists have tracked reef community composition across Belize for over 50 years, mostly in short-term, longitudinal studies [e.g., 11, 48, 50–52]. Belize has an extensive, 30-plus-year-old MPA network [48, 53] and a history of frequent large-scale disturbances, including bleaching events, disease outbreaks, hurricanes, and even an earthquake (Table 1). We surveyed fore-reef benthic communities at 15–18 m depth at 15 sites along the BBR during the summer months in 1997, 1999, 2005, 2009, and 2016 (Fig 1, Table 2). Due to logistical and resource constraints, only three of the 15 sites were surveyed every year: Bacalar Chico, Middle Caye, and Tacklebox (Table 2). Study sites were selected to maximize spatial heterogeneity and spanned a range of protections or management zones, including the Bacalar Chico, Hol Chan, and Glovers Reef Marine Reserves [5, 49]. They included five sites within fully protected (FP) zones (marine reserves), where only non-extractive activities are permitted, three sites within general-use (GU) zones, where fishing is permitted with some gear restrictions (e.g., prohibitions on longlines, gillnets, and spear-fishing with SCUBA) and modest fishing limits (e.g., catch-size limits for queen conch and lobster), and seven sites in unprotected (NP) zones, where fishing is allowed [48] (Table 2). Enforcement of fishing regulations in FP and GU sites was variable over time and ranged from inadequate to good (Table 2). Note that national seasonal closures for some species (e.g., Nassau grouper) and bans (e.g., on catching parrotfishes) applied to all three management zones.

### Benthic surveys

Benthic surveys were conducted *in situ* using SCUBA. At each site, dive teams laid out four to ten, 25–30 m x 2 m belt transects down the centers of reef spurs, perpendicular to the shoreline. The transects generally began on or near the shoulders of the spurs (i.e., beginning of the slope) at 15–18 m depth, shoreward of the drop-off that characterizes most of the reefs in Belize, and ran upward toward the reef crest. Transects were parallel to each other and were usually separated by >10 m. Divers worked in buddy pairs, in which one diver laid out the transect tape and the other used a digital camera in an underwater housing to obtain videos or still-frame images of the benthos. At each site, we videotaped or photographed the belt

**Table 1. Timeline of major disturbances to the Belizean Barrier Reef.**

| Year | Disturbance | References |
|---|---|---|
| 1980s | Acroporid-specific white-band disease | [58] |
| 1983 | *Diadema*-specific disease | [31] |
| 1998 | Temperature-induced coral bleaching | [58, 74] |
| 1998 | Hurricane Mitch | [75] |
| 2001 | Tropical cyclone Iris | [107] |
| 2000s | Yellow band disease | [108, 109] |
| 2005 | Temperature-induced coral bleaching | [21, 98–100] |
| 2007 | Hurricane Dean | [110] |
| 2009 | Earthquake | [111] |

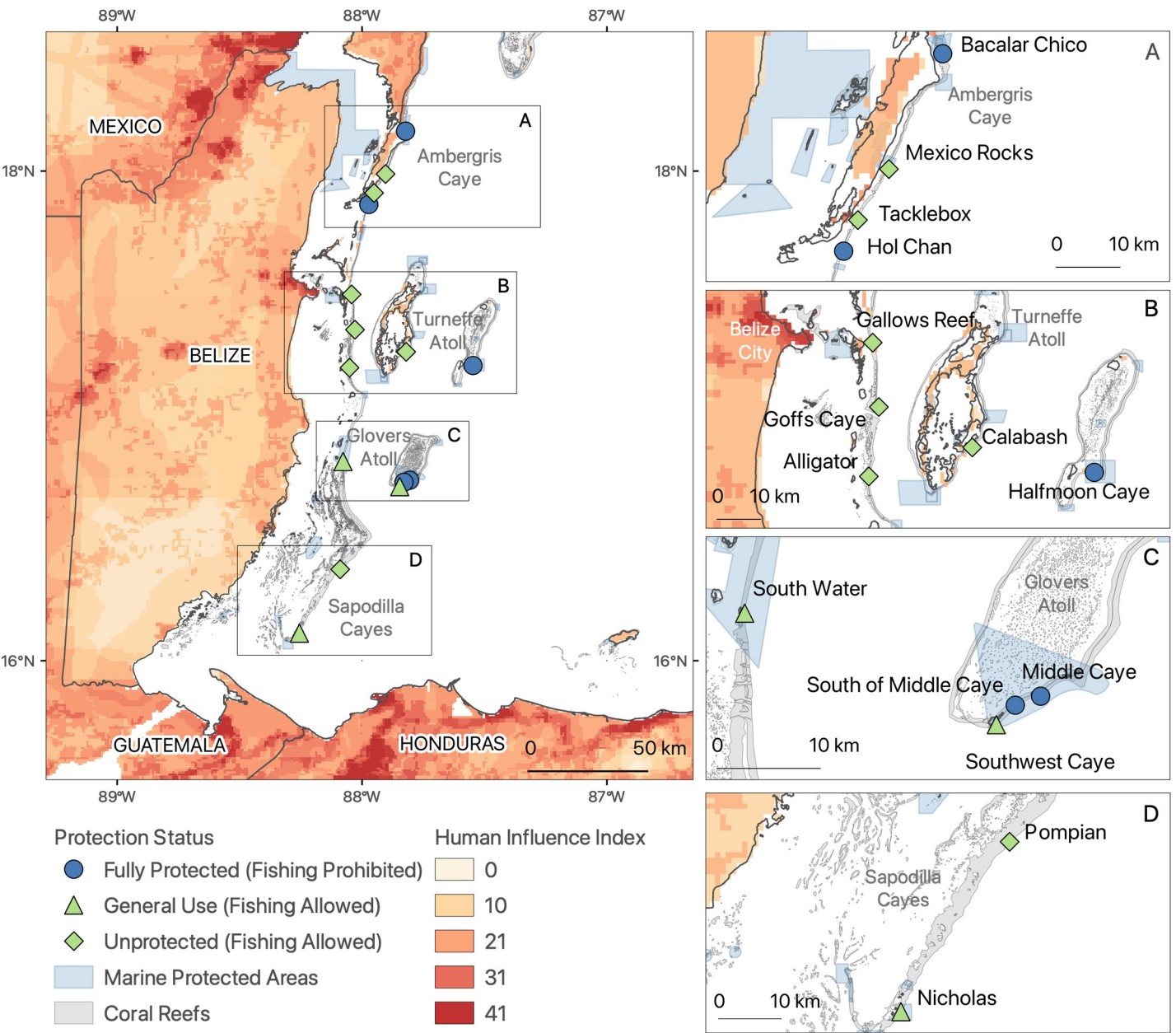

**Fig 1. Study sites along the Belizean Barrier Reef.** Sites are categorized by management level (fully protected, general use, unprotected) and prescribed protection status (fishing prohibited or allowed). Fishing is allowed in general use and unprotected sites (green) and is prohibited in fully protected sites (blue). Light blue polygons indicate the extent of MPAs. The Human Influence Index (HII), estimated at 1-km resolution for 1995–2004 across the landscape adjacent to study sites, is indicated in shades of light orange to red. GADM source: Database of Global Administrative Areas, Version 3.6 available at www.GADM.org.

transects at a standard distance of 25 cm above the benthos, using a bar projecting from the front of the camera housing to maintain the distance from the bottom. We also surveyed for *Diadema antillarum* from 2009 onward, but they were generally absent or extremely rare at all sites, possibly due to their refuge-seeking behaviors during the daytime when our surveys were completed [54]. In all sampling years except 2016, we obtained underwater videos along the belt transects and extracted still frames from those videos (as outlined below). In 2016, we photographed the transects using a GoPro HERO4 camera by swimming at a rate of 5–7 minutes

**Table 2. Site-related variables.**

| Site | 1997 | 1999 | 2005 | 2009 | 2016 | Latitude (°N) | Longitude (°W) | Protection Status | Fishing | Enforcement | HII 100km | HII 75km | HII 50km |
|------|------|------|------|------|------|--------------|---------------|-------------------|---------|-------------|-----------|----------|----------|
| Alligator | | | | Y | Y | 17.19660 | -88.05115 | NP | Allowed | None | 186234 | 100838 | 37768 |
| Bacalar Chico | Y | Y | Y | Y | Y | 18.16282 | -87.82222 | FP | Prohibited | M, M, M | 265863 | 113113 | 36727 |
| Calabash | | Y | | Y | | 17.26147 | -87.81970 | NP | Allowed | None | 124826 | 50174 | 4088 |
| Gallows Reef | | Y | | Y | Y | 17.49592 | -88.04255 | NP | Allowed | None | 222640 | 107300 | 41029 |
| Goffs Caye | | Y | | Y | | 17.35190 | -88.02880 | NP | Allowed | None | 196845 | 100781 | 37581 |
| Halfmoon Caye | | Y | | Y | Y | 17.20560 | -87.54679 | FP | Prohibited | M, M, G | 47605 | 3299 | 2480 |
| Hol Chan | Y | Y | Y | | Y | 17.86343 | -87.97238 | FP | Prohibited | G, G, G | 270893 | 118928 | 39311 |
| Mexico Rocks | Y | | | Y | Y | 17.98782 | -87.90382 | NP | Allowed | None | 266313 | 118363 | 32687 |
| Middle Caye | Y | Y | Y | Y | Y | 16.73700 | -87.80540 | FP | Prohibited | M, M, G | 82761 | 28341 | 100 |
| Nicholas | | Y | | Y | | 16.11230 | -88.25590 | GU | Allowed | I, I, M | 387947 | 166189 | 28520 |
| Pompian | | Y | | Y | | 16.37310 | -88.08910 | NP | Allowed | None | 228362 | 66688 | 9416 |
| South of Middle Caye | | Y | | Y | | 16.72880 | -87.82870 | FP | Prohibited | M, M, G | 87915 | 31021 | 432 |
| South Water Caye | | Y | | Y | | 16.81350 | -88.07760 | GU | Allowed | I, I, M | 150663 | 74522 | 30206 |
| Southwest Caye | Y | | Y | | Y | 16.71080 | -87.84610 | GU | Allowed | M, M, G | 92221 | 32447 | 1003 |
| Tacklebox | Y | Y | Y | Y | Y | 17.91060 | -87.95080 | NP | Allowed | None | 270394 | 120568 | 34625 |

Sites surveyed in a given year are represented with a Y, blanks are years when surveys did not occur. Latitude (Lat) and longitude (Lon) are expressed in decimal degrees. Protection refers to one of three of the following management regimes: (1) fully protected (FP) zones, where only non-extractive activities were permitted; (2) general-use (GU) zones with restrictions placed on certain fishing gear, total allowable catch limits, and seasonal closures; and (3) non-protected (NP) zones, where fishing was allowed [48]. Fishing was also allowed in GU, although prohibited in FP sites. Enforcement level was estimated for fully protected sites in 2011, 2014, and 2016 based on Eco-Audits, a qualitative analysis on enforcement and compliance with regulations performed by local MPA managers every 2–3 years. Sites with good (G) enforcement have regular patrols and overall satisfactory compliance. Sites with moderate (M) enforcement have regular patrols, but some poaching occurs and legal outcomes are insufficient. At sites with inadequate (I) enforcement, patrols are irregular, poaching persists, legal outcomes are insufficient, and the local communities express high levels of concern. Each site has three levels of enforcement (G, M, or I) corresponding to the years that an Eco-Audit was performed. Human Influence Index (HII) was estimated as the cumulative value within the 100-km, 75-km, and 50-km radii from the survey site.

along the 30-m-long transect and taking a photograph every five seconds. For photographs, we used the wide angle and 4:3 ratio setting of the GoPro camera, which corresponds to ~16–30 mm focal length and results in a frame area of ~0.25 m$^2$.

## Image extraction and analysis

Because of changes in imaging technology and analytical software over the course of this study, we used several techniques to extract and analyze the benthic images from the underwater transects. For sampling year 1997, we recorded Hi-8 video of each transect using two 30-watt ultrabright lights for illumination, whereas in 1999 and 2005 we used Sony 3chip mini DVR without illumination. We randomly selected 50 frames per transect, processed the images by de-interlacing, sharpening, and enhancing them, and saved them onto CD-ROMs. In 2009 and 2016, we switched to digital video. For 2009, we extracted the images from the video at a rate of 1-fps using Adobe Premiere Pro CC 2014. We ran the images through the Automator program in OS-X software to select every third, fifth or seventh image, depending on the length (in time) of the transect. We analyzed 15 images/transect/site for 2009 and 2016 because we found that we could obtain a similar level of inference about community composition with 15 images per transect as with the 50 images per transect suggested by Aronson et al. [55]. To select the images, we automated the process using a code in R version 3.6.3 to randomly choose, copy, and paste 15 images into a new folder from our source-folder of all images.

We analyzed the benthic cover of images from 1997–2005 using Coral Point Count software [56], and from 2009 and 2016 using CoralNet [57]. We manually input species-level benthic identifications for each of 10 random points overlaid onto each image [55]. When species-level identifications were not possible, benthic components were identified to genus or family (S1 Table in S1 File). All benthic components identified were pooled into five major benthic categories for analysis: (1) hard corals, including all scleractinian corals and milleporine fire corals; (2) macroalgae, including calcareous, filamentous, corticated and/or leathery algae; (3) crustose–turf–bare space (abbreviated CTB), which represents substrate that is bare, dead, covered in fine turf algae, and/or crustose coralline algae [50, 58]; (4) gorgonians; and (5) sponges (S1 Table in S1 File). Other minor categories such as invertebrates, bacterial mats, sediment, and rubble were also identified (S1 Table in S1 File). The corals *Orbicella annularis*, *O. favelota*, and *O. franksii* were pooled as *Orbicella* spp. because the species complex was not partitioned into the three species during the 1997 and 1999 data collection and because they were difficult to distinguish in some video frames (S1 Table in S1 File). In all instances, image-level point-count data were converted to percent-cover estimates for each transect, and we calculated the overall mean percent covers of each category for each site and year.

## Estimation of local impacts

We estimated the site-specific magnitude of local human impacts using the Global Human Influence Index (HII, version 2) from NASA's Socioeconomic Data and Applications Center (SEDAC) database [59]. The HII is a global dataset of 1-km grid cells aggregated from 1995–2004 designed to estimate location-specific human influence and thus potential impacts to natural populations and ecosystems via local direct and indirect human activities (e.g., harvesting and pollution). It is based on nine global data layers including human population density, land use and infrastructure (including land use/cover and nighttime lights), and access (which is estimated from coastlines, roads, railroads and navigable rivers). These aspects of human communities demonstrably predict local human impacts in many natural systems including coral reefs [6, 7, 28, 60–63]. We extracted HII values for the BBR (Fig 1) and calculated the sum of the HII scores of grid cells within a 50-km, 75-km, and 100-km buffer from the center-coordinates of each study site (S1 Fig in S1 File, Table 2). We used HII scores within the 50-km buffer for the final analysis because this metric performed well in exploratory models and it has been used successfully in prior work [5, 56]. We then tested whether this index of local human impacts was related to observed changes on the monitored benthic reef communities.

## Ocean temperature anomalies

Our measure of ocean-heatwave events was the site-specific frequency of Thermal Stress Anomalies (TSA Freq), obtained from the Coral Reef Temperature Anomaly Database (CoRTAD, Version 6) [64, 65] (S2 Fig and S2 Table in S1 File). TSA Freq is defined as the number of deviations of 1 degree Celsius (°C) or greater from the maximum weekly climatological sea-surface temperature during the 52 weeks preceding a reef survey. We also estimated the frequency of two related thermal stress metrics: (1) historical TSAs (TSA_Freq_hist), which is the number of times since the beginning of the dataset (1982) that TSA was $\geq 1°C$; and (2) the frequency of TSAs between survey years (TSA_Freq_btw_surveys), which is the number of instances since the previous survey year that TSA was $\geq 1°C$ (S1 Table in S1 File). In the final models, we used TSA Freq as the best metric to test for the effect of thermal stress on benthic groups because it performed better in exploratory models (S3 Table in S1 File). Other studies have found that TSA Freq is a significant predictor of coral-cover loss and coral-disease prevalence [66–68]. The CoRTAD is based on 4-km-resolution weekly measurements made by the

Advanced Very High-Resolution Radiometer (AVHRR) sensor (Pathfinder 5.0 and 5.2) beginning in 1982. Daytime and nighttime data were averaged weekly using data with a quality flag of 4 or better.

## Data analyses

To analyze changes in benthic composition and test for the effects of potential drivers of change, we built generalized linear mixed models (GLMMs) in a Bayesian setting using the *blme* package [69]. The response variables were the logit-transformed site- and year-specific percent covers of five key benthic categories (hard corals, macroalgae, CTB, gorgonians, and sponges), several coral taxa (genera, species, or groupings of coral species), and three major macroalgal functional groups (calcareous, fleshy, and corticated) (S1 Table in S1 File). The final models had year, protection status ("fishing prohibited," which included the sites within FP zones, and "fishing allowed" which included GU and NP sites), HII at the 50-km buffer, and TSA Freq as fixed effects; and Site as a random effect. A *blme* prior with a *wishart* distribution was imposed over the covariance of the random effect and modeled coefficients. We tried a *blme* (Bayesian Linear Mixed-Effects model) prior with different distributions such as *invwishart*, *gamma*, *invgamma*, and *null*, but they did not improve model performance. In exploratory analysis we modeled the interaction between TSA Freq and protection status as well as TSA Freq and HII; however, these interactions did not improve model fit and were not significant, so we dropped them from the models (see R Code in the GitHub repository "calves06/Belizean_Barrier_Reef_Change"). Thus, in the final models, all predictor variables were additive, and the REML estimation was used to fit the data as it provides unbiased estimates for the variance components. Prior to fitting models, we rescaled and centered all numerical fixed effects to optimize comparisons among variables. The average cover of each benthic category was finally modelled with a random-intercept model described as:

$$Logit\ (benthic\ cover)_{ij} =$$
$$\beta_1 + \beta_2 \times Year_{ij} + \beta_3 \times Protection\_Status_{ij} + \beta_4 \times HII{-}50km_{ij} + \beta_5 \times TSA\_Freq_{ij} +$$
$$\alpha_i + \varepsilon_{ij}$$
$$\alpha_i \sim N(0,\ \sigma_1{}^2)$$
$$\varepsilon_{ij} \sim N(0,\ \sigma_2{}^2)$$

where *Logit (benthic cover)*$_{ij}$ is the logit-transformed cover of each benthic category in the survey year $j^{th}$ ($j$ = 1997, . . ., 2016) of site $i^{th}$ ($i$ = 1, . . ., 15), $\beta_1$ is the intercept, and $\beta_2$ –$\beta_5$ are the coefficient estimates for each covariate (e.g., year, protection status, HII at 50 km, and TSA Freq). The term $\alpha_i$ is random intercept for site, which allows for random variation of the intercept $\beta_1$, and is assumed to be normally distributed (*N*) with mean *0* and variance $\sigma_1{}^2$. The term $\varepsilon_{ij}$ was the within-site variance of each benthic-group cover among years and is also assumed to be normally distributed with mean 0 and variance of $\sigma_2{}^2$.

We evaluated collinearity among fixed factors by assessing variance-inflation factors and chose a threshold of >3 to determine correlated variables. Correlated variables were dropped from the models. We tested for homoscedasticity (homogeneity of variances across predictor variables) by plotting residuals against fitted values. Comparing fitted and residual values suggested that our models were reasonable models of the means. We also examined the marginal and conditional R-squared values of the models.

To examine changes in community composition of all benthic taxa within sites and across years, we constructed a non-metric multidimensional scaling (NMDS) ordination using the *vegan* package in R. We used the Bray–Curtis dissimilarity index to calculate distances among

taxon-level cover data because it is robust to the large numbers of zeros (which denote absences) commonly found in ecological data and does not consider shared absences as being similar [70]. To determine the effects of covariates (year, TSA Freq, HII_50km, and protection status) on community-composition changes of benthic taxa we ran a Permutational Multivariate Analysis of Variance (PERMANOVA) using the Bray–Curtis dissimilarity index to calculate distance matrices. All statistical analyses were performed in R version 3.6.3. The code and processed data are available at https://github.com/calves06/Belizean_Barrier_Reef_Change.

### Ethics statement

The field research was performed under permits from the Belize Fisheries Department to MM, NB, KC, CF, CC, and JFB, including permit numbers 000018–09 and 000028–11.

### Results

HII varied by >400-fold among sites (Table 2) and was the greatest at the sites closest to the most altered human landscapes and lowest at the most geographically isolated reefs (Fig 1 and S1 Fig in S1 File). For example, HII was high around Belize City, much lower in southern Belize, and high in Honduras. HII values were very high at the sites near Belize City, where local impacts such as murky water flowing out of the Belize River are obvious, including Gallows and Alligator (values were 41029 and 37768, respectively; Fig 1, Table 2). HII values for sites near the rapidly developing Ambergris Caye resorts were also high, including Hol Chan and Bacalar Chico marine reserves (39311 and 36727, respectively; Table 2). In contrast, the geographically isolated sites, where water clarity was far better and local impacts from the major human developments of the mainland should have been lower, had far smaller HII values (e.g., 100 and 2480 for Middle and Halfmoon Cayes, respectively). These low-HII reefs, geographically isolated from most local stressors, acted as controls for reefs closer and presumably more impacted by human development.

Among the main benthic groups of interest—hard corals, macroalgae, CTB, gorgonians, and sponges—we observed a significant decline in hard-coral and CTB cover, significant increases in macroalgal and gorgonian cover, and no significant change in sponge cover (Fig 2, S3 Fig in S1 File, Table 3). Protection status (fishing allowed *versus* prohibited) was not predictive of observed spatiotemporal variation in hard-coral, macroalgal, CTB, or sponge cover (Figs 2 and 3, S4 Fig in S1 File, Table 3) and was marginally related to gorgonian cover. HII was also unrelated to hard-coral, macroalgal, CTB, or sponge cover but was significantly and negatively related to gorgonian cover (Fig 3, Table 3). TSA Freq, our metric of ocean-heatwave frequency, was significantly and negatively related to the cover of hard corals and gorgonians, and unrelated to the cover of macroalgae, CTB, and sponges (Figs 3 and 4). Total coral cover and the cover of four coral taxa—*Acropora* spp., *Orbicella* spp., *Montastrea cavernosa*, and *Porites* spp.—were negatively related to heatwave frequency (Figs 3 and 4). Local protection within MPAs or geographic isolation from local impacts (sites with low HII scores) did not reduce the effect of ocean-temperature anomalies on these four affected coral taxa: the TSA Freq*HII and TSA Freq*Protection Status interaction terms were not significant (see R Code in the GitHub repository "calves06/Belizean_Barrier_Reef_Change" and S3 Table in S1 File). In fact, the model structures with the interaction terms generally performed worse (they had higher AIC scores) than the additive models, and thus these interaction terms were dropped from the final models.

The observed overall decline in hard-coral cover across the Belizean Barrier Reef from 26.3% (± 7.3 SD) in 1997 to 10.6% in 2016 (± 3.5 SD; Fig 2, S4 Fig in S1 File) was driven by changes in the abundance of a handful of reef-building coral species (Fig 5). Notably, there was

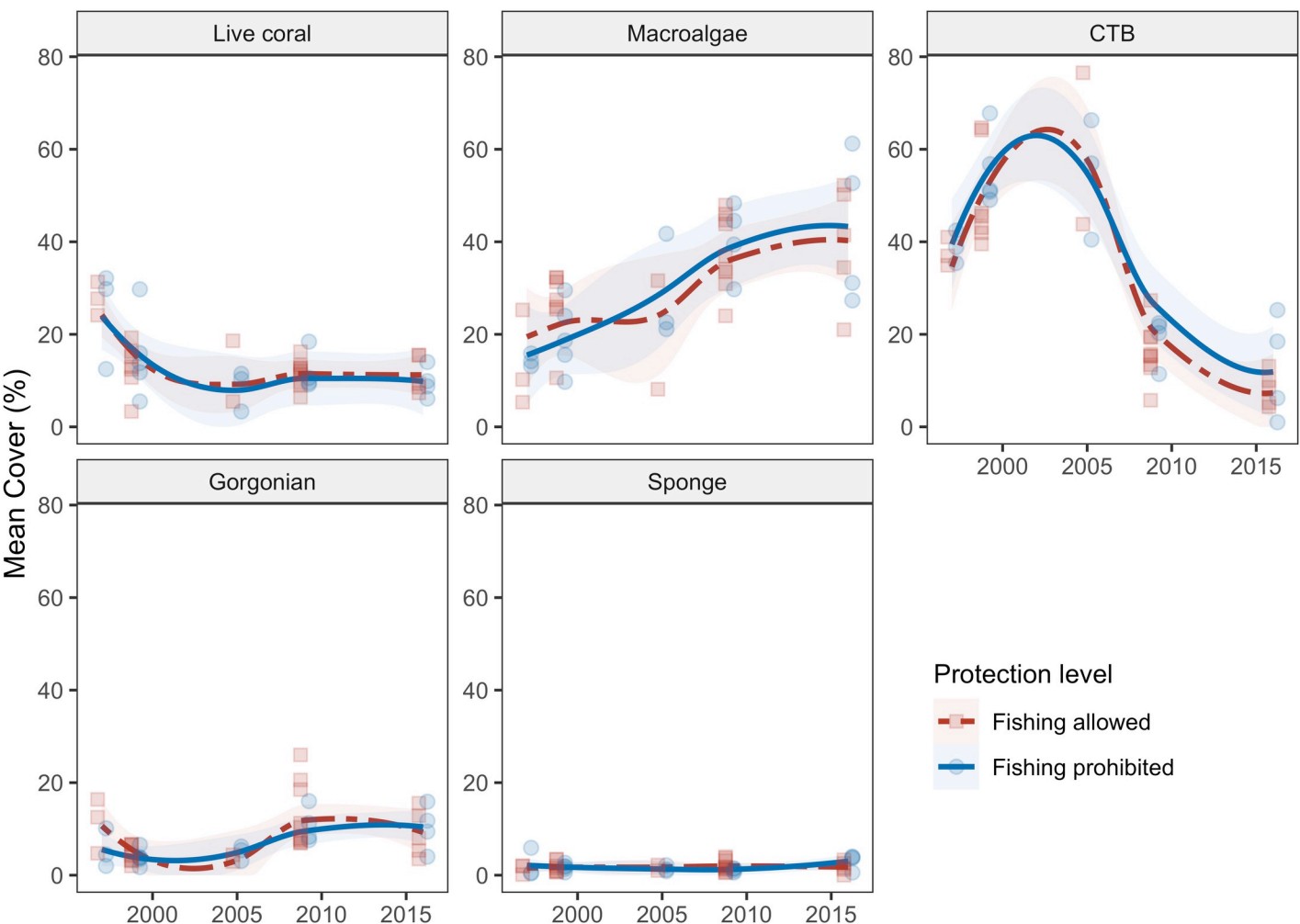

**Fig 2. Percent cover of five benthic categories over time grouped by protection status.** Points are site means, lines are loess-smoothed curves with a span of 1, and shading indicates the 95% confidence intervals of the loess fits.

a significant decline of *Orbicella* spp., with mean cover falling from 12.7% (± 7.4 SD) in 1997 to 2.2% (± 0.9 SD) in 2016 (Fig 5, S4 Table in S1 File; model estimate = - 0.719, p < 0.001). This decline was predominantly observed from 1997 to 1999, which included a major bleaching event and Hurricane Mitch (Fig 5, Table 1), and from 2005 to 2009, which included a second bleaching event, and Hurricane Dean (Fig 5, Table 1). The cover of hard-coral taxa such as *Acropora* spp., *Colpophyllia natans*, and the combined cover of 'other' coral taxa (e.g., *Mycetophyllia* spp., *Madracis* spp., and *Favia fragum*; see S1 Table in S1 File for a complete list) also declined significantly from 1997 to 2016 (Fig 5, S4 Table in S1 File). The cover of the coral taxa *Agaricia agaricities*, *Diploria/Pseudodiploria* spp., *Montastraea cavernosa*, *Siderastrea* spp., *Porites astreoides*, and branching *Porites* spp. (*P. porites*, *P. furcata*, and *P. divaricata*) remained relatively low and did not change significantly during the study period (Fig 5, S4 Table in S1 File). The cover of *Agaricia tenuifolia* slightly but significantly increased (Fig 5, S4 Table in S1 File). Protection status and HII were not significant predictors of spatial and temporal changes of any coral taxa (S4 Table in S1 File), except for *P. astreoides*, for which sites with higher cover had greater HII (S4 Table in S1 File). The cover of *Acropora* spp., *M. cavernosa*, *Orbicella* spp.

**Table 3. Estimated regression parameters for the coverage of benthic groups.**

| Benthic group/Terms | Estimates | Std. error | F-statistic | p-value | Sig. |
|---|---|---|---|---|---|
| *Hard Coral* | | | | | |
| (Intercept) | -1.877 | 0.126 | -14.849 | < 0.001 | *** |
| Year | -0.492 | 0.103 | -4.773 | < 0.001 | *** |
| Fishing vs. No Fishing | 0.127 | 0.225 | 0.567 | 0.570 | |
| HII at 50 km | 0.324 | 0.217 | 1.492 | 0.136 | |
| TSA Freq | -0.383 | 0.117 | -3.278 | 0.001 | ** |
| *Marginal $R^2$/Conditional $R^2$* | *0.347/ 0.665* | | | | |
| **Macroalgae** | | | | | |
| (Intercept) | -0.929 | 0.148 | -6.274 | < 0.001 | *** |
| Year | 0.925 | 0.108 | 8.576 | < 0.001 | *** |
| Fishing vs. No Fishing | 0.138 | 0.265 | 0.522 | 0.602 | |
| HII at 50 km | 0.307 | 0.256 | 1.202 | 0.229 | |
| TSA Freq | 0.225 | 0.123 | 1.829 | 0.067 | |
| *Marginal $R^2$/Conditional $R^2$* | *0.480/0.775* | | | | |
| **CTB** | | | | | |
| (Intercept) | -0.998 | 0.149 | -6.695 | < 0.001 | *** |
| Year | -1.622 | 0.194 | -8.360 | < 0.001 | *** |
| Fishing vs. No Fishing | 0.212 | 0.248 | 0.853 | 0.394 | |
| HII at 50 km | -0.243 | 0.243 | -0.999 | 0.318 | |
| TSA Freq | 0.255 | 0.205 | 1.245 | 0.213 | |
| *Marginal $R^2$/Conditional $R^2$* | *0.613/0.652* | | | | |
| **Gorgonian** | | | | | |
| (Intercept) | -2.152 | 0.097 | -22.092 | < 0.001 | *** |
| Year | 0.407 | 0.111 | 3.658 | < 0.001 | *** |
| Fishing vs. No Fishing | -0.373 | 0.166 | -2.240 | 0.025 | * |
| HII at 50 km | -0.450 | 0.162 | -2.777 | 0.005 | ** |
| TSA Freq | -0.352 | 0.121 | -2.921 | 0.003 | ** |
| *Marginal $R^2$/Conditional $R^2$* | *0.430/0.550* | | | | |
| **Sponge** | | | | | |
| (Intercept) | -3.443 | 0.159 | -21.724 | < 0.001 | *** |
| Year | -0.209 | 0.172 | -1.215 | 0.224 | |
| Fishing vs. No Fishing | 0.070 | 0.273 | 0.258 | 0.797 | |
| HII at 50 km | -0.125 | 0.265 | -0.469 | 0.639 | |
| TSA Freq | -0.299 | 0.188 | -1.591 | 0.112 | |
| *Marginal $R^2$/Conditional $R^2$* | *0.099/0.329* | | | | |

Shown are the estimated regression parameters, standard errors, F-statistics, p-values, significance levels, and *marginal/conditional $R^2$* from the final Bayesian generalized linear mixed models for each benthic group. Significance levels (Sig.) are: *** < 0.001; ** < 0.01; * < 0.05.

*Porites* spp., and 'other' coral taxa were negatively correlated with TSA frequency (Fig 5, S4 Table in S1 File).

The temporal dynamics of macroalgae varied substantially among functional groups (Fig 6). The average cover of calcareous macroalgae (e.g., *Halimeda* spp.) remained relatively low at 2.3% (±3.0 SD) across sites (Fig 6). This group was unaffected by protection and local impacts, but showed a slight but significant decline over time, likely due to higher cover in some sites in 1999, which was also positively related to higher thermal-stress anomalies (S5 Table in S1 File). From 1997 to 2016, the average cover of fleshy macroalgae (e.g., *Dictyota* spp.) doubled from

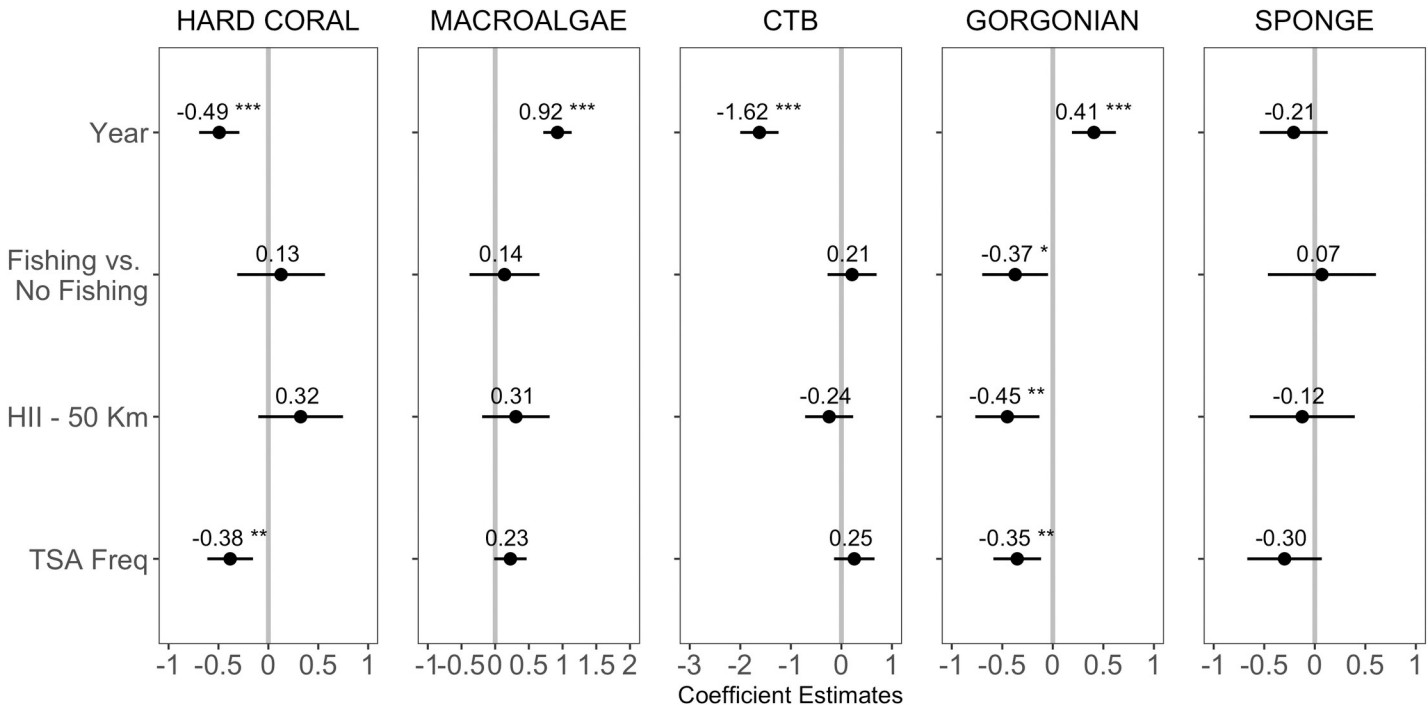

**Fig 3. Effect-sizes (± 95% CI) of covariates from the Bayesian generalized linear mixed-effect model on the five benthic groups.** Values above points are effect sizes. CIs crossing the vertical grey line represent non-significant effects. Significance levels: *** = 0.001; ** = 0.01; * = 0.05.

12.8% (± 8.0 SD) to 25.2% (± 13.4 SD) and was positively associated with higher local impacts (albeit weakly; p < 0.047), but protection and heat waves had no effect (Fig 6, S5 Table in S1 File). Similarly, the cover of corticated macroalgae (e.g., *Lobophora variegata*) increased significantly to 13.0% (mean ± 11.6 SD), and up to 42% in some sites such as the Hol Chan Marine Reserve (Fig 6, S5 Table in S1 File). In contrast, protection and local human impact had no effect on corticated macroalgae, but lower cover was found in sites with more thermal-stress anomalies (S5 Table in S1 File).

Based on the ordination analysis in Fig 7, there were major compositional shifts in the dominant benthic assemblages during 1997–2005 (left) and 2009–2016 (right) at every site (Table 4). The PERMANOVA showed that, among all covariates, time explained about 50% of the variability in benthic community changes (F = 45.8, p < 0.001) and was the only significant predictor of change in overall community composition (Fig 7, Table 4). Protection status, HII, and TSA frequency combined only accounted for 6% of community differences and were not good predictors of overall change of all taxa studied (Table 4). In 1997–2005, the benthic communities of the BBR were dominated by CTB and long-lived, massive reef-building corals such as *Orbicella* spp. and *C. natans*. During 2009–2016, composition had shifted to domination by small and/or weedy hard-coral species, macroalgae, and gorgonians (Fig 7).

## Discussion

### Temporal patterns of change

During our 20-year study of the dynamics of fore-reef benthic communities along the BBR, we documented declines in the reef-building corals *Acropora cervicornis*, *A. palmata*, and *Orbicella* spp., and large increases in the cover and biomass of fleshy and corticated seaweeds including *Dictyota*, *Lobophora*, *Turbinaria*, and *Sargassum*. Our results are concordant with

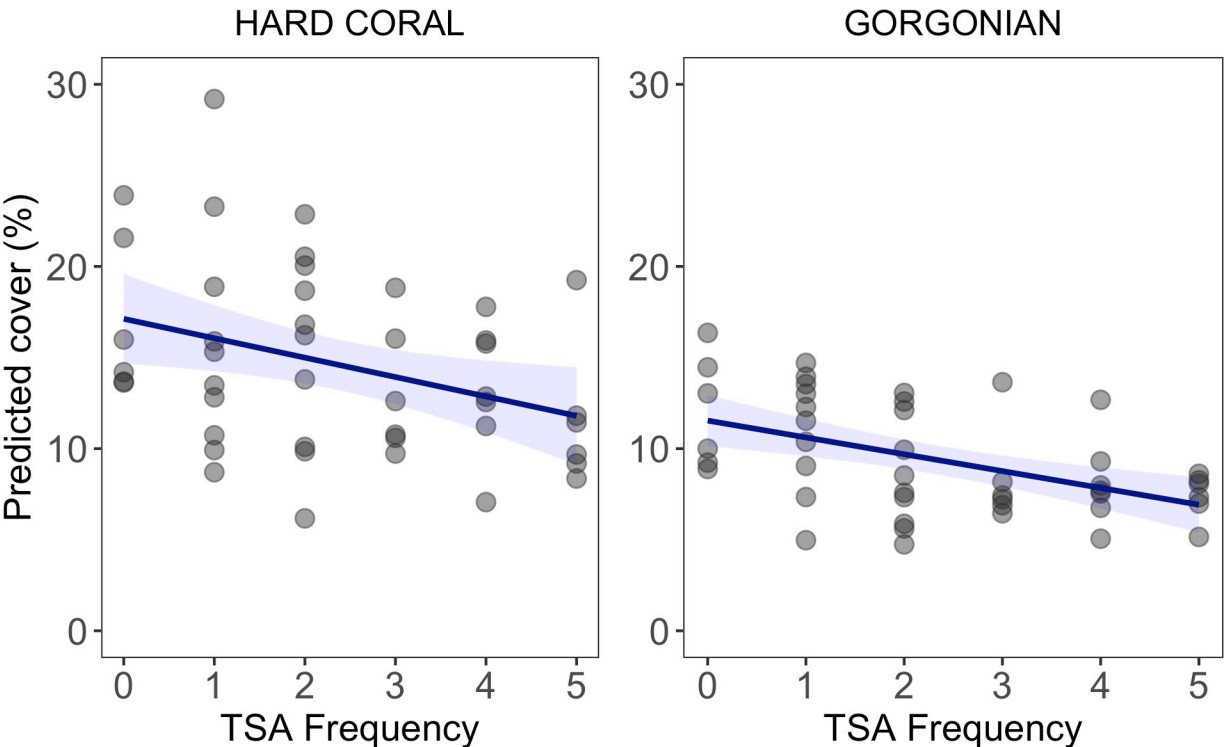

**Fig 4. Effect of TSA frequency on hard-coral and gorgonian cover.** Points are predicted benthic-group cover (back-calculated from logit transformation) from a Bayesian generalized liner mixed model accounting for time, protection status, and HII. Blue lines are the fitted curves of the models and shaded areas are the 95% CIs.

previous studies in Belize that documented similarly striking shifts in hard-coral and macroalgal cover [71]. For example, the patch reefs of Glovers Reef Atoll had ~80% hard coral and 20% fleshy-macroalgal cover in 1970–1971 but had shifted to 20% hard-coral and 80% macroalgal cover by 1996–1997 [71]. A longitudinal study of *A. palmata* along the Mexican portion of the Mesoamerican Barrier Reef also reported declines in acroporids, with *A. palmata* decreasing from 7.7% in 1985 to 2.9% in 2012 [72]. Prior to the beginning of our study, acroporid abundance had already declined across much of the BBR due to both hurricanes and white-band disease [11, 72, 73]. Most remaining *A. cervicornis* and *A. palmata* colonies were killed by high ocean temperatures during the 1998 mass-bleaching event [58, 74].

Corals declined in the first few years of our study (1997–1999), then remained relatively constant at both protected (~11.8 ± 1.5%, fishing prohibited) and unprotected sites (~11.9 ± 0.9%, fishing allowed; Fig 2). In fact, this general stasis in coral cover has been apparent across the region, especially on low-coral-cover reefs, for the last several decades [2, 75]. Given the frequent disturbances on the BRR during this period (Table 1), stability in coral cover is technically evidence of "resilience"[76]. However, what remained in 2016 were low-coral-cover reefs (mean coral cover within sites ranged from 6.1–15.6%), made up almost entirely of species tolerant of frequent, acute disturbances and longer-term environmental changes (Fig 5). We do not see this as good news, but rather as the inevitable ecological outcome of the replacement of functionally important, framework-building taxa (especially *Acropora* spp. and *Orbicella* spp.) by the weedy species favored in the Anthropocene disturbance regime [77]. Regardless, the arrival of stony coral tissue loss disease on the BBR in 2017 and its subsequent spread may very likely end this period of low-coral-cover stasis by reducing cover even further.

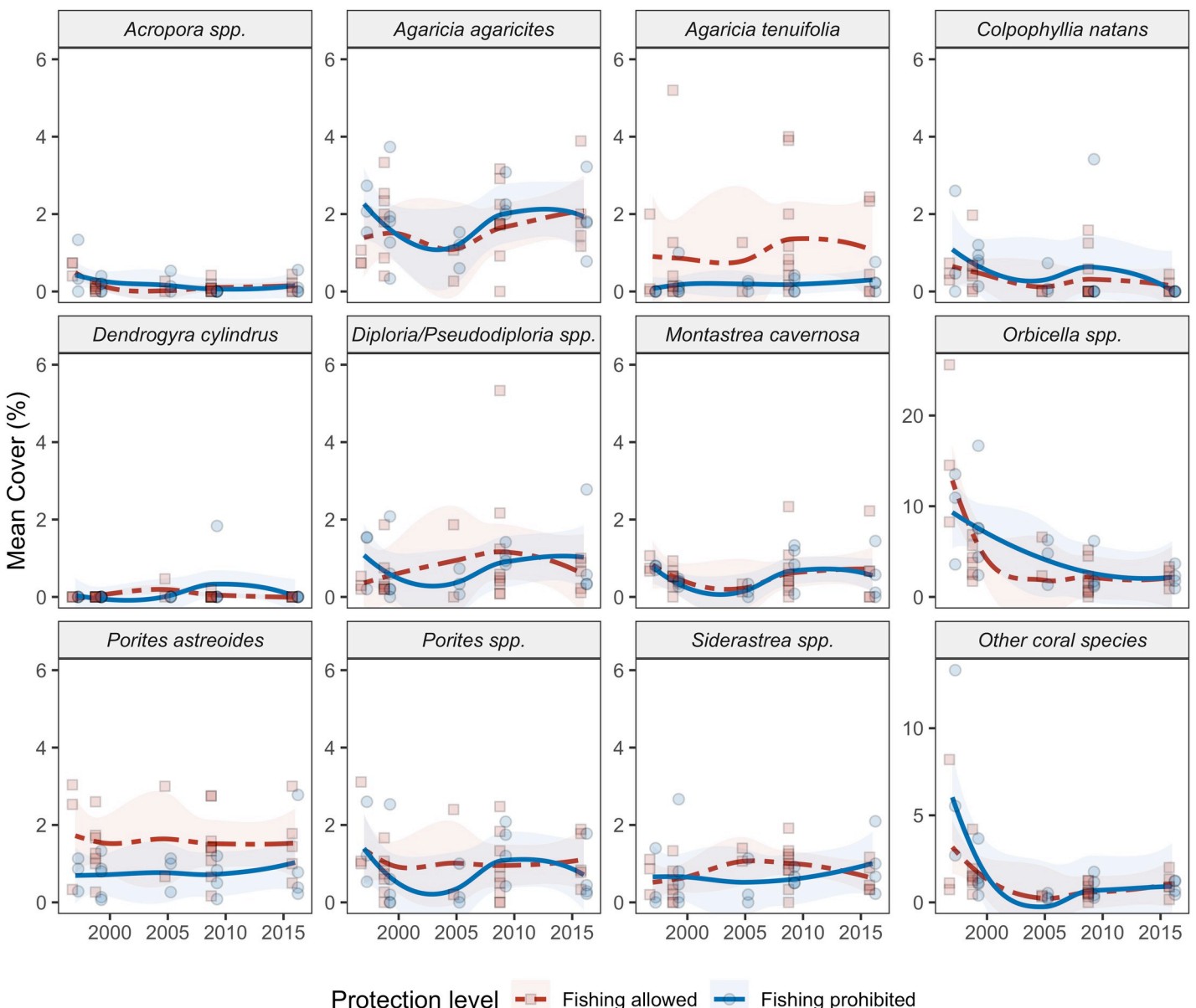

**Fig 5. Mean percent cover of twelve taxonomic categories of hard corals, grouped by protection status.** Fishing occurs at sites coded red and is prohibited at sites coded blue. Points are site means for each surveyed year, lines are a loess smoothed curves with a span of 1, and shading indicates the 95% confidence intervals of the loess fits.

We found that the benthic composition changed over time and benthic assemblages were ecologically distinct between the earlier and later sampling intervals (1997–2005 and 2009–2016; Fig 7). For instance, the hard corals *Acropora* spp. and *Orbicella* spp. were more often present and more dominant (both had higher relative and absolute cover) in the early sampling years. In contrast, fleshy macroalgae and gorgonians came to dominate during later sampling years. The cover of 'weedy' coral taxa such as *Porites* spp. and *Agaricia* spp. remained relatively consistent throughout the course of the study (Fig 5).

Shifts in the dominant benthic taxa have been documented across the Caribbean and are often linked to regional disturbances such as herbivore declines, coral diseases, and mass-

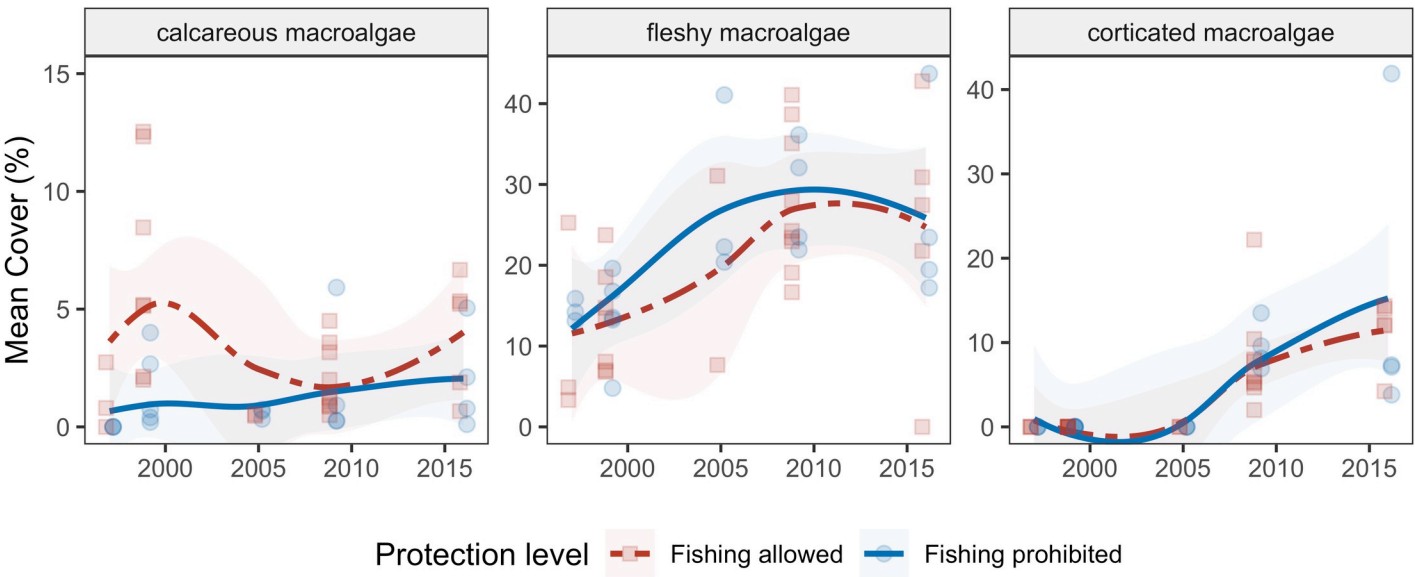

**Fig 6. Mean percent cover of three functional categories of macroalgae, grouped by protection status.** Points are site means for each surveyed year, lines are loess smoothed curves with a span of 1, and shading indicates the 95% confidence intervals of the loess fits. For the algal genera in each group see S4 Table in S1 File; for model results see S5 Table in S1 File.

bleaching events [2, 3, 78–80]. Across seven subregions in the Caribbean, Schutte et al. [2] found significant declines in hard-coral cover and increases in macroalgal cover from 1970–2005. Corals failed to recover in the Florida Keys [81] and the U.S. Virgin Islands [82] due to subsequent, repeated disturbances. The coral reefs of Bonaire exhibited similar trends over 15 years of bleaching, storms, and diseases, with a 22% decline in coral cover and an 18% increase in macroalgal cover by 2017 [83]. These trends were also apparent in our study.

## Effects of local protection

The primary management response to the degradation of coral reefs has been the implementation of MPAs [34, 41, 44, 84]. Within well-designed and enforced MPAs, fish abundance, biomass, and diversity often increase and in some cases spill over into adjacent, non-protected areas [73, 85–88]. MPAs can reduce other extractive activities that could directly or indirectly impact coral populations [89]. However, a large majority of studies have found that MPAs are not slowing or preventing the decline of reef-building corals [52, 67, 81, 90–93], particularly in response to large-scale disturbances. A recent meta-analysis of 18 studies, encompassing 66 MPAs, reported that MPAs did not affect coral loss or recovery in response to large-scale disturbances including disease, bleaching, and storms [39]. Our results for the BBR agree with this broad consensus.

Belize's network of protected areas, designed and implemented in part to prevent the degradation of benthic reef assemblages on the BBR, has not achieved this goal. Our results complement previous findings for Belize reporting the failure of individual MPAs or the network overall to protect and restore populations of overharvested reef fishes [5, 46, 89][but see [73]]. We documented a statistically and ecologically significant decline in hard-coral cover, an increase in macroalgae and gorgonians, and a substantial decline of CTB, regardless of protection status (Fig 2). Similar coral declines in isolated, well-protected, and seemingly 'pristine' locations have been documented at dozens of other sites globally [80, 94].

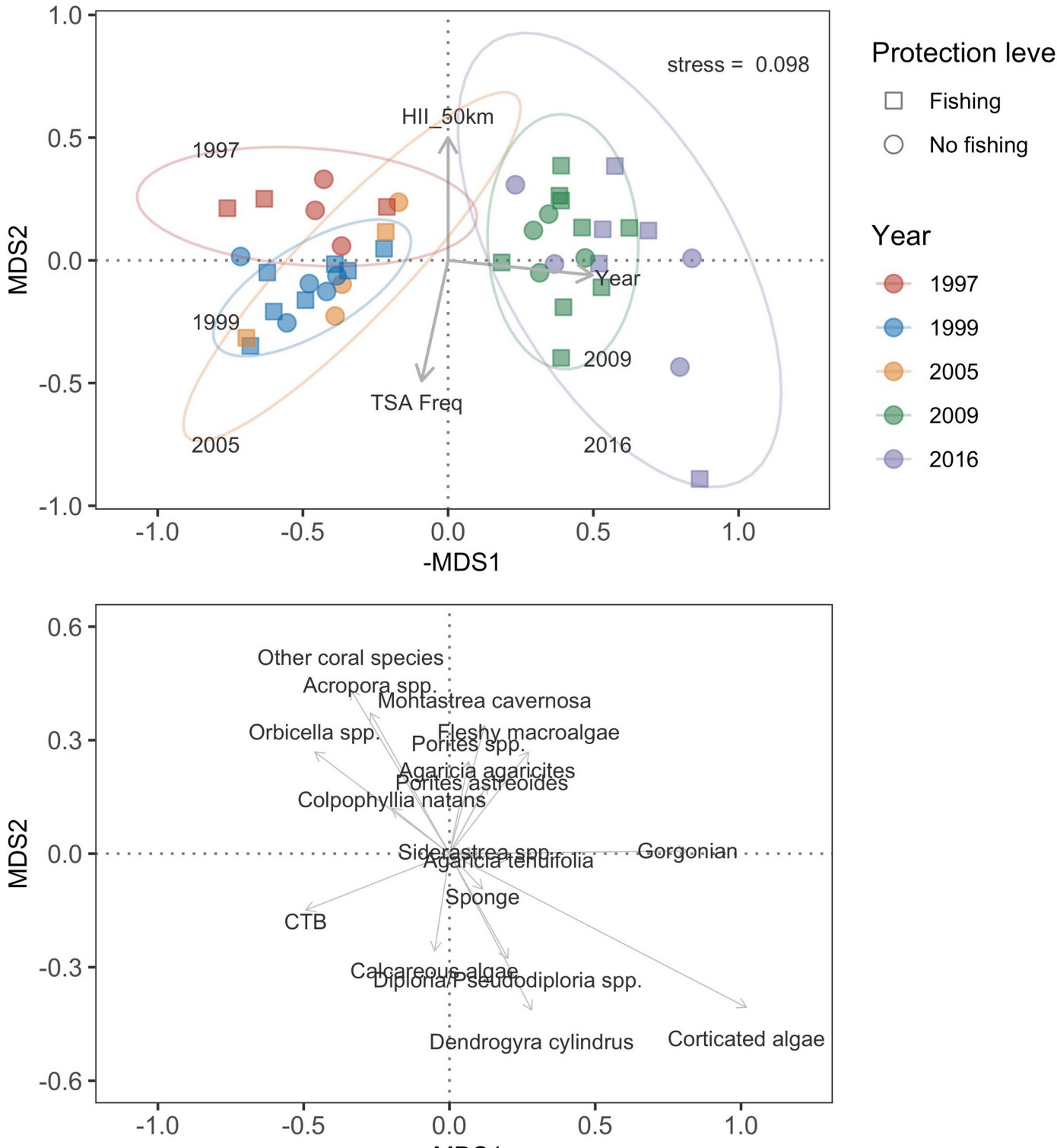

**Fig 7. Non-metric multidimensional scaling (MDS) plot depicting taxon-level cover data.** In the top panel, points represent individual sites, circles are fishing sites, and squares are no-fishing sites. Data are colored by year. Arrows represent the fitted loadings scores for Year, TSA_Freq, and HII_50km. In the bottom panel, the arrows and labels represent the loadings of specific benthic categories loadings. The Bray–Curtis dissimilarity matrix was used and the stress value was 0.098.

**Table 4. Results of the Permutational Multivariate Analysis of Variance (PERMANOVA) using the bray-curtis dissimilarity index to determine the effects of covariates in changing the composition of benthic communities, based on cover.**

| Term | df | SS | $R^2$ | F | Pr(>F) | Sig. |
|------|----|----|----|---|--------|------|
| Year | 1 | 2.347 | 0.501 | 45.761 | <0.001 | *** |
| HII 50km | 1 | 0.137 | 0.029 | 2.679 | 0.062 | |
| TSA Freq | 1 | 0.072 | 0.015 | 1.401 | 0.218 | |
| Protection status | 1 | 0.075 | 0.016 | 1.465 | 0.200 | |
| Residual | 40 | 2.051 | 0.438 | | | |
| Total | 44 | 4.683 | 1.000 | | | |

Significance level (Sig.): *** < 0.001.

df: degrees of freedom, SS: sum of squares.

The most likely explanation for why MPAs had no measurable effect on most benthic community components, and in particular no effect on corals, is that they simply are not designed to mitigate the primary agents that are currently responsible for reef degradation [95]. One parameter that well-implemented fisheries restrictions *can* influence is the individual sizes and population densities of herbivorous fishes—and consequently the biomass and cover of macroalgae [34, 42, 96]. This, in theory, could benefit coral populations, primarily by promoting recruitment, at least of the weedy coral taxa coming to dominate coral reefs [33, 34].

Enforcement of the protected areas in our study varied among sites and over time from 'good' (regular patrols and minimal poaching, as in Hol Chan) to 'inadequate' (irregular patrols and poaching, as in South Water) (Table 2). Cox. et al. [48] monitored reef fish biomass from 2009 to 2013 within eight protected areas (all included in this study) and eight control sites on the BBR and found no general effect of local protection on fish biomass, including herbivorous parrotfishes, snappers, or groupers. One exception was Half Moon Caye; among the most isolated and well-protected reefs in Belize's MPA network (Table 2), where predatory-fish biomass was among the highest in the BBR [5]. The primary reasons for the poor performance of many MPAs in the region appear to be illegal fishing, poor compliance with fishing regulations, and absence of sufficient enforcement [97] (Table 2).

McClanahan and Muthiga [73] found that the fully protected marine reserve on Glover's Reef Atoll, which encompassed two of our sites, had strongly positive effects on the biomass of carnivorous fishes including snappers and groupers, but no effect on parrotfishes. In fact, parrotfish biomass declined at both the protected and control sites (n = 4) during the 22-year study. McClanahan and Muthiga [73] concluded, "Fisheries regulations are unlikely to rapidly restore hard corals on these patch reefs, which have slowly transitioned to algal dominance since first described in 1970." Their conclusion is concordant with this and other studies on Glover's Atoll and across the BBR generally [48, 73, 93].

## Effects of ocean heating and other large-scale disturbances

We suspect that changes in the benthic assemblages of coral reefs along the BBR are due primarily to the large-scale disturbances to the system over the last several decades, including mass-bleaching events in 1998 and 2005 caused by anthropogenic climate change, disease outbreaks, and seven hurricanes that may have been exacerbated by climate change (Table 1). The decline in *Orbicella* spp. (Fig 5) was likely due primarily to mortality from coral bleaching in 1998 [58, 74] and 2005 [21, 98–100] and yellow-band disease in the early 2000s (Table 1).

Anthropogenic climate change was clearly a significant driver of the dramatic shifts in community composition that occurred on the BBR over the two-decade study. The absolute covers of four common coral taxa (*Acropora* spp. *Orbicella* spp., *Montastraea cavernosa*, and *Porites*

spp.) were negatively related to heatwave frequency (Figs 3 and 4). This result is in agreement with other studies that have documented coral mortality and consequent declines in coral cover following the temperature-induced mass-bleaching events on the BBR in 1998 and 2005 [50, 58]. Many other studies have documented the role of ocean heatwaves in coral decline around the world [21, 22, 67, 80, 91, 92, 101–104].

### Effects of local development and subsequent stressors

Human population growth in Belize and the dramatic increase in tourism infrastructure, including on the isolated cays along the BBR, have likely altered the physical and chemical conditions of nearshore marine habitats, especially within the lagoons adjacent to some of the most drastically altered shorelines. We measured the effects of coastal impact by testing whether cumulative human influence scores for the land, including the cays, within a 50-km radius of a given reef was related to the benthic community dynamics at that site during our 20-year study. More intense local development and land use change should be positively associated with the magnitude of localized stressor such as pollution, including sediments, nutrients, herbicides, etc. on the adjacent coastal waters. We assumed that the influences of the stressors originating onshore or at the land–sea interface should dissipate with isolation from the source, assuming they are point-source stressors.

Our results suggest that the local impacts had no measurable effect on hard-coral cover: there was no association between HII and change in the cover of any coral species. HII was, however, significantly and negatively related to changes in gorgonians and positively associated with the cover of *Porites astreoides*. In addition, the cover of fleshy macroalgae was weakly and positively associated with sites of higher HII (S5 Table in S1 File). There is abundant evidence that local impacts, including pollution, fishing, and coastal land-use practices, can severely impact coral populations [28, 29, 105, 106] and promote the growth of fleshy macroalgae [37, 56, 101]. Yet, even when these stressors are clearly present, they are often overwhelmed by the effects of large-scale disturbances including ocean heatwaves and storms [20, 39, 61, 71]. There is no doubt that local human activities in Belize are affecting shallow, lagoonal environments, but that may not be true of the deeper, fore-reef environments that were the focus of our study. It is also possible that HII is a poor predictor of the local human stressors on adjacent marine habitats, although the positive correlation of fleshy macroalgae and HII suggests the contrary. Unfortunately, time-series of direct measurements of sediment flux, nutrient pollution, turbidity, etc. are not available for the BBR (nor for most reefs globally, but see [105]). The absence of fine-grained, time-series data on environmental parameters potentially influenced by human land use has greatly hampered tests of their role in reef decline. Although, there is no doubt these factors pose a threat, and in some cases they have been shown to influence local reef dynamics [106], making strong inferences about their impact at a specific location is essentially impossible in the absence of such data.

### Conclusion

Our data show a substantial shift in the state of coral reefs along the Belizean Barrier Reef over a two-decade period rife with large-scale disturbances. We documented declines in the key reef-building coral genera *Acropora* and *Orbicella*, subsequent increases in macroalgal and gorgonian cover, and an overall change in the benthic assemblages over the two-decade study. Ocean-heatwave frequency was a significant predictor of coral-population declines over time, whereas local protection and local human impacts had few measurable effects on benthic taxa. However, environmental changes caused by local human activities, such as increased nutrient concentrations, are not monitored in Belize, making it challenging to assess directly their

effects. The rapid elimination of global greenhouse emissions is clearly paramount for the survival and recovery of the BBR. In tandem with such efforts, we urge local authorities to increase resources to support the enforcement of existing MPAs and to mitigate any possible effects of coastal development on Belize's coral reefs.

## Supporting information

**S1 File.**
(DOCX)

## Acknowledgments

We thank the many volunteers who assisted with data collection, logistical support and image analysis over the years. We are extremely grateful for our partnerships with staff from the Belize Fisheries Department, Belize Coastal Zone Management Project, Wildlife Conservation Society, Marine Research Center at the University of California–Berkeley, Belize Audubon Society, Pelican Beach Resort, Rum Point Inn, Sea Sports Belize, Healthy Reefs for Healthy People, Bacalar Chico National Park and Marine Reserve, Hol Chan Marine Reserve, The Nature Conservancy, Southern Environmental Association, Toledo Institute for Development and Environment, and the Smithsonian Institution.

## Author Contributions

**Conceptualization:** Abel Valdivia, Richard B. Aronson, Nadia Bood, Karl D. Castillo, Clare Fieseler, Melanie McField, John F. Bruno.

**Data curation:** Catherine Alves, Abel Valdivia, Nadia Bood, Karl D. Castillo, Courtney Cox, Clare Fieseler, Zachary Locklear, Melanie McField, John F. Bruno.

**Formal analysis:** Catherine Alves, Abel Valdivia, Nadia Bood, Clare Fieseler, Zachary Locklear, Melanie McField, James Umbanhowar.

**Funding acquisition:** Richard B. Aronson, Nadia Bood, Courtney Cox, Clare Fieseler, Melanie McField, John F. Bruno.

**Investigation:** Catherine Alves, Abel Valdivia, Nadia Bood, Karl D. Castillo, Courtney Cox, Clare Fieseler, Melanie McField, Laura Mudge, John F. Bruno.

**Methodology:** Abel Valdivia, Richard B. Aronson, Nadia Bood, Karl D. Castillo, Clare Fieseler, Zachary Locklear, Melanie McField, John F. Bruno.

**Project administration:** Catherine Alves, Abel Valdivia, Nadia Bood, Courtney Cox, Clare Fieseler, Melanie McField, John F. Bruno.

**Resources:** Karl D. Castillo, Courtney Cox, Clare Fieseler, Melanie McField, John F. Bruno.

**Software:** Melanie McField.

**Supervision:** Richard B. Aronson, Courtney Cox, John F. Bruno.

**Visualization:** Catherine Alves, Abel Valdivia.

**Writing – original draft:** Catherine Alves, Abel Valdivia, John F. Bruno.

**Writing – review & editing:** Catherine Alves, Abel Valdivia, Richard B. Aronson, Nadia Bood, Karl D. Castillo, Courtney Cox, Clare Fieseler, Melanie McField, Laura Mudge, John F. Bruno.

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
