## [Decision Letter · Decision Letter 0]

26 Apr 2021

PONE-D-21-07319

Twenty years of change in benthic communities across the Belizean Barrier Reef

PLOS ONE

Dear Dr. Bruno,

Thank you for submitting your manuscript to PLOS ONE. After careful consideration, we feel that it has merit but does not fully meet PLOS ONE’s publication criteria as it currently stands. Therefore, we invite you to submit a revised version of the manuscript that addresses the points raised during the review process.

We look forward to receiving your revised manuscript.

Kind regards,

Loren D. Coen, Ph.D.

Academic Editor

PLOS ONE

Journal Requirements:

[This manuscript is based upon work supported by the National Science Foundation (DGE-1650116 to CA, OCE-0940019 to JFB, and partial support from OCE-1535007 to RBA), the Rufford Small Grant Foundation, the National Geographic Society, the International Society for Reef Studies/Center for Marine Conservation Reef Ecosystem Science Fellowship, the Elsie and William Knight, Jr. Fellowship from the Department of Marine Science at the University of South Florida, the J. William Fulbright program, the Organization of American States Fellowship, the World Wildlife Fund-Education for Nature Program, the Kuzimer-Lee-Nikitine Endowment Fund, the Nicholas School International Internship Fund at Duke University, the Lazar Foundation, and the Environment, Ecology and Energy Program, the Department of Biology, and the Chancellor’s Science Scholar Research Fund at the University of North Carolina at Chapel Hill. Any opinions, findings, and conclusions or recommendations expressed in this material are those of the authors and do not necessarily reflect the views of the funders. The funders had no role in study design, data collection and analysis, decision to publish, or preparation of the manuscript.].    

We note that one or more of the authors are employed by a commercial company: ECS Federal, Inc.

3. We note that Figure 1 and S1 in your submission contain map images which may be copyrighted. All PLOS content is published under the Creative Commons Attribution License (CC BY 4.0), which means that the manuscript, images, and Supporting Information files will be freely available online, and any third party is permitted to access, download, copy, distribute, and use these materials in any way, even commercially, with proper attribution. For these reasons, we cannot publish previously copyrighted maps or satellite images created using proprietary data, such as Google software (Google Maps, Street View, and Earth). For more information, see our copyright guidelines: http://journals.plos.org/plosone/s/licenses-and-copyright.

You may seek permission from the original copyright holder of Figure 1 and S1 to publish the content specifically under the CC BY 4.0 license. 

If you are unable to obtain permission from the original copyright holder to publish these figures under the CC BY 4.0 license or if the copyright holder’s requirements are incompatible with the CC BY 4.0 license, please either i) remove the figure or ii) supply a replacement figure that complies with the CC BY 4.0 license. Please check copyright information on all replacement figures and update the figure caption with source information. If applicable, please specify in the figure caption text when a figure is similar but not identical to the original image and is therefore for illustrative purposes only.

Additional Editor Comments:

Comments to Author (Highlights to Focus On)

Please carefully follow the recommendations of reviewers 2-4, as well as my comments and concerns with regard to this manuscript. The reviewers have made many constructive comments, edits, and also suggestions regarding a number of potential concerns associated with the paper as submitted. I think that is can be revised, and then resubmitted for re-review if all or most of the aforementioned concerns, omissions (Methods especially), and caveats, etc. are addressed. Curious how much this paper overlaps with Cox et al. 2017 (MEPS 563)?

If you are prepared to undertake the work required, I would be pleased to reconsider my decision. If the Authors believe that they cannot be addressed then a detailed line by line rebuttal is required.

**
Abstract
**

I wonder if the title might be modified to better reflect the actual emphasis corals vs. benthic communities? The paper really elaborates sufficiently on inverts and macroalgae to suggest replacement? The detailed values in the abstract are not sufficiently reflected in the Results and Discussion. Storms and bleaching not evaluated directly either. Please elaborate later in paper. The Abstract needs to be edited to read more clearly (e.g., L42-45). Please also describe for uninitiated value of HII (local vs. regional scale and also how generated, Line 45). Lines 45-46, only indirect estimate of local protection or fishing (L48 also). Line 49-50 tone down this statement (Reviewer #1).

Reviewers have concerns as to the global conclusions the authors focus on in both the Abstract, Introduction and Discussion (though each are slightly different).

See reviewer #4 as to concerns regarding ‘bold statements based on 15 sites…”

Mention the massive development in country and especially coast and cayes.

**Introduction**

Reviewer #4 mentions concerns over dismissing of local efforts given the concerns about the paper. In Intro as well as Discussion address more explicitly caveats both here with the study approaches, but also given other efforts worldwide dealing with similar concerns.

Shoreline protection not discussed or relevant here. Mention removal of herbivores also (Line Lines 63 on, Hay and others). Lines 308-310) “We attribute changes in the benthic assemblages of coral reefs along the BBR primarily to the large-scale disturbances to the system over the last several decades, including seven hurricanes and two mass-bleaching events caused by anthropogenic climate change (Table 1)” yet hurricanes not evaluated directly here (Line 66) nor the other major disturbance events mentioned in table (see also Aronson, Precht, Macintyre, others). Are loss of herbivores, smothering, and competition with macroalgae only secondary drivers (Lines 75-79)? Many would disagree.

Belize has undergone unbelievable development in the past 30 years. Main threats to the its forests include massive expansion of agriculture, housing, and tourism, the latter is exploding both on the mainland, as well as throughout the offshore cayes. The country has incredibly high deforestation rates, poor management of wastes, very rapid coastal development (new airport in works), increasing poverty, very weak institutional and legal govt institutions, more recently discovery of oil will impose tremendous threats to its health and well-being into the future.

Agriculture (citrus, sugar cane, bananas, etc.), and aquaculture have increased tremendously along with major declines in the country’s forests. Belize has a deforestation rate >2x that of other Central American countries. Over 17% of its forests have been lost from 1980-2010. Riparian deforestation is even greater at over 5x that of its uplands.)! More recently forest cover in Belize has decreased from nearly 73% in 1989 to 61.64% in 2012. Some estimate that by 2020 nearly 60% will have been lost, and by 2040 all will be gone (Belize Govt, 2005). Obviously, its forests provide soil stabilization estuaries and reduces the runoff while supporting some of CA most diverse forest communities.

It’s obvious that worldwide corals are declining significantly from both natural and human-related stressors. These worldwide declines are resulting in major shifts in community dominance from hard corals to macroalgal, sponge, or even gorgonians. The roles of increasing algal cover on coral reefs has been well documented (McCook et al., 2001; Hay et al., Mumby et al. numerous citations), but as the others all compete with corals the mechanisms are not easily teased apart by observations and inference. Mention in more detail some of the other drivers such as nutrient loading (Line 78, Reviewer 3) and citations. Potential conceptual model might be worth adding (Reviewer 3)?

Seaweed blooms (i.e.* Sargassum*) no citations (line 84). How much driven by explosion of development, improper treatment of sewage, sand scavenging for mangrove shoreline fill (e.g., Ambergris), runoff and removal of mangroves on cayes offshore, and deforestation? Much more complex system than even the coral folks suggesting here. They underplay this perhaps because loss of herbivores and detailed documentation of macroalgal community quite absent from analyses here.

The relevant citations and ‘stories’ provided appear to be somewhat biased (Reviewer #1 how debate framed) in support of the author’s ultimate conclusions that local efforts (e.g., MPAs), and other management actions are having little or no impact on stemming the decline of Belizean Barrier Reef benthic communities based on their definitions and approaches. They conclude that the loss of stony corals (only one segment of the diverse benthic plant and animal community) was primarily related to increasing ocean temperatures, while being largely unaffected by any local efforts (e.g., MPAs) as estimated by their metric, HHI.

Perhaps because of the time-frame no mention is made of for example, stony coral tissue loss disease (SCTLD) a potentially novel disease now impacting corals across the Caribbean while appears not being exacerbated by elevated temperatures.

Additionally, perhaps because of the limited assessment at all of the macroalgal community (as well as related ‘turfs’ and coralline algae) communities which are often precursors to developing macroalgal domination where herbivory is limited (Lewis 1986, Lewis et al. 1987), invasive or exploding *Sargassum* sp. is not even mentioned. See comments at end of Reviewer #1 review, I agree on lack of macroalgal cover as function of MPA status. See Fig. 2 comments and related observations.

Overfishing on offshore cayes occurred as early as the 1980s, and *Diadema* never really returned to their post-dieoff levels either (Line 78-79) though occurring much earlier than this descriptive study. Without an in depth analysis of the major components (species) replacing or competing with corals (macroalgae, etc.) the paper understates a large benthic component, while doing little to expand on this in the Discussion or for that matter in the Methods as to why this major community element is lumped into just % cover?

My concerns are that the paper focuses on direct estimates of the dominant corals at a limited number of sites and footprints (transects) through time while using indirect broadscale estimates that have either a great deal of error at the scale measured here, or are fraught with problems (are the MPAs used here indeed adequately protected from fishing?).

Lines 95-98 “We found that benthic-community composition changed substantially during this period, and that the observed loss of corals was negatively related to ocean heatwaves and largely unaffected by local impacts, fishing or protection status” dismissed local efforts without an objective assessment of the rigor of Belizean MPAs. Our concern (myself and reviewers #1 & 4) is that the conclusions might lead Belize and other countries to give up managing their resources rather than more strictly enforcing or adequately researching and developing rigorous metrics that sufficiently estimate fishing pressure, management of reserves, potential options for more rigorous enforcement, restoration, etc.

**M & M and Results**

Line 103-104, “Belize has an extensive, 30-plus-year-old MPA network (46)”. Anyone working there knows that the resource management of offshore areas poor at best (cf. overfishing of shrimp in 1980s, lobster and conch fisheries, etc.). Stating that 5 sites were fully protected (reserves) is unfortunately not true (Line 110). Modest fishing is similarly a stretch and removal of nursery areas means that recruitment throughout the area significantly impacts reef fish reproduction. Furthermore SAV is disappearing across many cayes in Belize smothered by extensive intertidal live and dead *Sargassum* (e.g., Ambergris Caye). How much did the transects vary N to S through time? State that 1 of the 2 divers laid down the tape (Line 124). Were there start and endpoints permanently established? Have no idea. Why use video vs. film or digital cameras in housings? Could you increase your sample size by analyzing more images on each videos? What was the estimated area of each image captured?

Only three of the 15 sites surveyed yearly. Why were the transects only carried out in the summer? They cite Aronson et al. (1994) for the number of quadrats to use, but in that paper, Aronson et al. say (Pg. 6) that macroalgal cover is maximal during summers and "seasonal changes within a site could change estimates of coral cover, as more or less living coral is obscured by the algae" (cf. Diaz-Pulido and Garzón-Ferreira 2002). They also say that "algal destruction by storms is maximal in the winter". Obviously their summer sampling is a temporal snapshot and may not reflect true coral cover at reef sites. Many algal efforts use a Random Point Contact sampling design (RPC). This approach is probably a better way to delineate coral cover when macroalgae is present, but it is obviously time consuming. State Lines 150-151, “We manually input species-level benthic identifications for each of 10 random points overlaid onto each image (51)”. Again why only ID corals? Lines 153-157, state “five benthic categories: (1) crustose–turf–bare space (or CTB)… bare, dead, covered in turf algae, and/or crustose coralline algae, (2) hard corals (which includes all scleractinian corals and milleporine fire corals), (3) macroalgae, including algae in the genus* Halimeda*, (4) gorgonians, and (5) sponges.” Why mention only *Halimeda* amongst the easily identifiable algae, *Lobophora*, etc. (cf. McCook et al. 2001, Box and Mumby 2007)? No differentiation beyond “macroalgae” for some reason? Not even to browns, reds, greens? How did you deal with seasonality of algae, etc. as well as layers of community components growing on or near live or dead coral rubble?

See reviewer #1 & 4 as to concerns regarding ‘bold statements based on 15 sites… over 19 years”. Recognize study’s limitations explicitly. Concerns as to estimating local drivers based on global datasets! Also, “quality of the HII dataset without independent assessments to approximate reality.” Assume that the fishing pressure, development, etc. as characterized by potentially inaccurate underestimate of human impacts using the HII index data, especially in underdeveloped countries. HII are purely land based so that offshore anthropogenic drivers on islands poorly estimated in model. MPAs can have positive effects on coral cover (see suggestions). No fish or urchin estimates as indirect assessment of mgmt. effectiveness (see reviewer #1 same concern!). “If authors have no independent data on MPA enforcement (reviewer #1) easy to poke holes in dataset, so acknowledge its limitations upfront.”

Reviewer #1 is most concerns with the “lack of independent confirmation that the fished no fished sites as designated were truly enforced” during the 19 years of the present study. Those of us that have worked there for extended periods know that Belize does not have the best record of enforcement be it finfish, lobster, conch or other! Proximity to developments or resorts is often a major driver although isolated cayes can have significant fishing pressure from non-Belizean fisherman.

The study site types were not equally apportioned N to S. It that because of the lack of for example MPAs in the south for example? I had BTW a hard time resolving the much of the information provided on the Figures in the main as well as the Suppl. sections.

Transects were 25-30 m x 2 m with depths beginning from 15-18 m depth shoreward and shallower. Only thing that did not shift was the distance of 25 cm (however, this distance can vary greatly given the type of substrate or community imaged and the lens type) from benthos. Quite confusing as to the various approaches employed across years from the non-digital video sampling in '95, '97, and '05 they used de-interlaced, sharpened, and enhanced 50 random frames per transect. Then in '09 and '16, they went to digital videos and decreased the sample size (state got a similar level of inference (based on what analysis?) about community composition with only 15 images per transect as compared to the 50 images per transect suggested by Aronson et al.). Fifteen frame grabs along a 25 m transect seems like a small number to characterize a transect? If the authors were not using permanent transects this haphazard transect placement along the center of spurs makes their small sample size (15 images per transect, and a range of from 4-10 transects per site) and 10 points per image insufficient. Can you explain the range of transects? Was it because of the size of the areas among sites?

Were the non-digital, enhanced images of similar quality to the digital images? For 2016, they used a GoPro Hero 4 not it appears from video but rather selected images every 5 sec. (L130-131). That device has a lot of curvature because of the fisheye lens. Were their GoPro images corrected in some way to adjust for the area across cameras?

Perhaps a table showing the changes through time of gear type, no. of transects (ranged from 4-10), number of images analyzed, depth of those images, etc. would help to reduce the confusion or insufficient provided detail? Size of the quadrat or field of view analyzed, estimated pixels, etc.

No mention of mangrove losses on the cayes offshore, but losses are considerable all related to human activity, most associated with tourism development in coastal areas that may not adequately be captured by the HHI estimates used here.

You used HHI database to estimate human influences. You might give distances here from shore for the 15 stations. You state (Lines 169-170) that HII uses nine global data layers including human population density, land use, and access (which are estimated from coastlines, roads, railroads and navigable rivers, etc.) but we are interested in islands (cayes) offshore potentially little impacted by these 9 layers? Reviewers #3,4 have some questions on thee scales for HII. Is 100 km, 75 km and 50 km resolution relevant to the scale of the cayes offshore or was that was available from the HII? In your approach did these larger scales weigh against selecting for regional to local factors. See Reviewer #4 who is very familiar with these databases and asked if there was any vetting of the data with local experts to see if the changes detected in the HII approximated reality?

Pay attention to Reviewer #3s concerns on logit-transf. % cover, Wishart, etc. Are year and TSA freq. autocorrelated or confounded? Additional questions concerns need to be addressed.

For TSA Freq. are they at a sufficiently fine scale to estimate temps at depths given that upwelling off the drop-offs (15-18 m) regularly brings cooler water into the shallower reef areas. Note that no indirect estimates of disease or stressors or reef coral condition were included to correlate with these presumed temperature anomalies.

Lines 228-229, “All statistical analyses were performed in R version 3.6.3. The code and processed data are available at https://github.com/calves06/BRC”. **However, the link provided was broken.**

**
Discussion (covered also above)
**

The Discussion seems quite abbreviated given the extent of the dataset and analyses. I assume it will grow after reading the comments and adding changes occurring in Belize during the sampling, related concerns and caveats, etc.

Line 300 for the first time brown algal genera are mentioned sort of in passing only in Discussion why (cf. Hoek et al. 1978, Ferrari et al. 2012)?

See reviewer #1 & 4 regarding concerns as to conclusions based on limited stations, etc.

Reviewer #1 “Need to better recognize the limitations of the study.” This should be done in the Abstract as well as the Discussion.

Reviewer #1 emphasizes in closing that the paper stresses linkage of heat waves and climate change (line 88), and coral (or for that matter urchin) diseases (Line 73) or not (line 353). They suggest that there is no conclusive link between heat stress events and disease events! However, the paper conflates that two. Dataset is over 5 yrs. old and SCTLD is ravaging reefs throughout the Caribbean with warming events playing no role! See paper’s recommendations at end of Abstract. They need to be revised given this and other concerns by reviewers and myself. Furthermore SAV is disappearing across many cayes in Belize smothered by extensive intertidal live and dead *Sargassum* (e.g., Ambergris Caye). Whether this is local eutrophication or Sahara sands is up in the air.

**
Tables, Figures, Appendices
**

Table 1 are these data or relevant info utilized in the analyses or Discussion ever?

I struggled to resolve much of the data provided on many of the Figures in the main, as well as the Suppl. sections. Suggest that Fig. 1 have sample sizes next to three potential ‘management’ levels. Also, use symbols next to the types of fishing levels vs. colors. Was there a site in Hol Chan? Adding Ambergris might help folks vs. site names only?

Figures 2, 4-6 hard to resolve the symbols on Figures relative to the legends. Use cross hatching or solid or open or something rather than slight shading.

Figure 5 we all know that there really is no such thing as fishing and no fishing levels. The lines look the same in b & w! Don’t use red and blue! Hatch one line. Same on Fig. 2.

Add A, B on Figure 6 vs. top and bottom panels.

Reviewer #1 wonders if the raw data will be provided anywhere? No link in S1

S5 why no similar list for other spp. like gorgonians or corallines or macroalgae at least?

**
Some Potential References?
**

Box, S.J., and P.J. Mumby, 2007. Effect of macroalgal competition on growth and survival of juvenile Caribbean corals. Mar. Ecol. Prog. Ser. 342:139–149.

Diaz-Pulido, G., and J. Garzón-Ferreira, 2002. Seasonality in algal assemblages on upwelling-influenced coral reefs in the Colombian Caribbean. Botanica Marina 45:284-292.

Ferrari, R., M. Gonzalez-Rivero, J.C. Ortiz, and P.J. Mumby, 2012. Interaction of herbivory and seasonality on the dynamics of Caribbean macroalgae. Coral Reefs 31:683-692.

Government of Belize, 2005. Forest Department’s Five Year Strategic Plan, 2005-2010. Ministry of Natural Resources, Local Government and the Environment.

Hoek, C. van den, A.M. Breeman, R.P.M. Bak, and G. van Buurt, 1978. The distribution of algae, corals and gorgonians in relation to depth, light attenuation, water movement and grazing pressure in the fringing coral reef of Curaqao, Netherlands Antilles. Aquat. Bot. 5:1-46.

Jompa, J., and K.J. McCook, 2002a. The effects of nutrients and herbivory on competition between a hard coral (Porites cylindrica) and a brown alga (*Lobophora variegata*). Limnology Oceanography 47:527-534.

Jompa, J., and K.J. McCook, 2002b. Effects of competition and herbivory on interactions between a hard coral and a brown alga. J. Exp. Mar. Biol. Ecol. 271:25–39.

Jompa, J., and K.J. McCook, 2003a. Contrasting effects of turf algae on corals: massive *Porites* spp. are unaffected by mixed species turfs, but are killed by the red alga *Anotrichium tenue*. Mar. Ecol. Prog. Ser. 258:79–86.

Jompa, J., and K.J. McCook, 2003b. Coral-algal competition: macroalgae with different properties have different effects on corals. Mar. Ecol. Prog. Ser. 258: 87-95.

Huffard, C.L., S. von Thun, A.D. Sherman, K. Sealey, and K.L. Smith, Jr., 2014. Pelagic *Sargassum* community change over a 40-year period: temporal and spatial variability. Marine Biology 161:2735–2751.

Lenes, J.M., J.M. Prospero, W.M. Landing, J.I. Virmani, and J.J. Walsh. 2012. A model of Saharan dust deposition to the eastern Gulf of Mexico. Marine Chemistry 134-135:1-9.

Lewis, S.M., 1986. The role of herbivorous fishes in the organization of a Caribbean reef community. Ecological Monographs 56:183-200.

Lewis, S.M., J.N. Norris, and R.B. Searles. 1987. The regulation of morphological plasticity in tropical reef algae by herbivory. Ecology 68:636-641.

Louime, C, J. Fortune, and G. Gervais, 2017. *Sargassum* invasion of coastal environments: a growing concern. Am. J. Environ. Sci. 13:58-64.

McClanahan, T.R., and N.A. Muthiga, 1998. An ecological shift in a remote coral atoll of Belize over 25 years. Environ. Conserv. 25: 122-130.

McCook, L.J., J. Jompa, and G. Diaz-Pulido, 2001. Competition between corals and algae on coral reefs: a review of evidence and mechanisms. Coral Reefs 19:400-417.

Magdaong et al., 2014; Mellin et al., 2016; Rogers, 2009; Selig and Bruno, 2010; Strain et al., 2019 (reviewer #4).

Milledge, J.J. and Harvey, P.J. 2016. Golden tides: problem or golden opportunity? The valorisation of *Sargassum* from beach inundations. J. Mar. Sci. Engin. 4:60.

Mumby, P.J., N.L. Foster Glynn, and E.A. Fahy, 2005. Patch dynamics of coral reef macroalgae under chronic and acute disturbance. Coral Reefs 24:681-692.

Norstrom, A.V., et al., 2009. Alternative states on coral reefs: beyond coral-macroalgal phase shifts. Mar. Ecol. Prog. Ser. 376:295–306.

Nugues, M.M., and R.P.M. Bak, 2006. Differential competitive abilities between Caribbean coral species and a brown alga: a year of experiments and a long-term perspective. Mar. Ecol. Prog. Ser. 315:75-86.

Puk, L.D., N. Cernohorsky, A. Marshell, J. Dwyer, K. Wolfe, and P.J. Mumby, 2020. Species-specific effects of herbivorous fishes on the establishment of the macroalga *Lobophora* on coral reefs. Mar. Ecol. Progr. Ser. 637:1–14.

Ramlogan, N.R., P. McConney, and H.A. Oxenford, 2017. Socio-economic impacts of *Sargassum* influx events on the fishery sector of Barbados. CERMES Technical Report 81. 90pp.

Ruyter van Steveninck, E.D. de, and R.P.M. Bak, 1986. Changes in abundance of coral reef bottom components related to mass mortality of the sea urchin Diadema antillarum. Mar. Ecol. Prog. Ser. 34:87-94.

Slattery, M., and M.P. Lesser, 2021. Gorgonians are foundation species on sponge-dominated mesophotic coral reefs in the Caribbean. Front. Mar. Sci. 8:654268.

Sotka, E.E., and M.E. Hay, 2009. Effects of herbivores, nutrient enrichment, and their interactions on macroalgal proliferation and coral growth. Coral Reefs 28:555-568

Vermeij, M.J.A., I. van Moorselaar, S. Engelhard, C. Hörnlein, D.M. Vonk, and P.M. Visser, 2010. The effects of nutrient enrichment and herbivore abundance on the ability of turf algae to overgrow coral in the Caribbean. PLoS ONE 5 (12): e14312.

Wang, M., and C. Hu, 2017. Predicting Sargassum blooms in the Caribbean Sea from MODIS Observations. Geophysical Research Letters 44:3265–3273.

Wang, M., C. Hu, B.B. Barnes, G. Mitchum, B. Lapointe, J.P. Montoya, 2019. The great Atlantic *Sargassum* belt 365:83-87.

Young, C., 2008. Belize’s ecosystems: threats and challenges to conservation in Belize. Tropical Conservation Science 1:18-33.

Reviewers' comments:

Reviewer's Responses to Questions

**Comments to the Author**

1. Is the manuscript technically sound, and do the data support the conclusions?

Reviewer #1: Partly

Reviewer #2: Partly

Reviewer #3: Yes

Reviewer #4: Yes

2. Has the statistical analysis been performed appropriately and rigorously? 

Reviewer #1: Yes

Reviewer #2: Yes

Reviewer #3: Yes

Reviewer #4: Yes

3. Have the authors made all data underlying the findings in their manuscript fully available?

Reviewer #1: Yes

Reviewer #2: Yes

Reviewer #3: Yes

Reviewer #4: Yes

4. Is the manuscript presented in an intelligible fashion and written in standard English?

Reviewer #1: Yes

Reviewer #2: Yes

Reviewer #3: Yes

Reviewer #4: Yes

5. Review Comments to the Author

Reviewer #1: Long data sets such as this are comparatively rare. While the original intent was not to publish a long term dataset when the project started, the data were collected for various reasons and it makes sense to present it as a long look back.

Reviewer #2: In this contribution, the authors address the long-running debate among coral reef ecologists of the relative importance of local vs. broad-scale stressors on reef health (here, as coral cover) with a time-series analysis of reef transects from sites in fished vs. MPA reefs across 15 sites on the BBR over 19 years. This reviewer is in complete agreement that broad-scale stressors are what matters, particularly for Caribbean reefs (although not in complete agreement with how the debate is framed in this paper, see below). I am generally supportive of publication of this work, particularly because standardized time-series data are valuable and rare. However, there are weaknesses in this paper that will probably not convince “the other side” in the debate (not that this could occur, even with incontrovertible proof), and I think the authors should consider revisions that better recognize the limitations in this study.

The biggest limitation is the lack of independent confirmation (validation) that MPAs were enforced during the study period. Figures indicate “fishing” and “no fishing” without anything to back this up. The easiest way to do this is to provide data on fish abundance or biomass, minimally parrotfish abundance or biomass, and then analyze the data relative to MPA effectiveness (as fish biomass). Unenforced MPAs are likely no different than fished areas, and Belize does not have a good record of MPA enforcement. Easy-access reefs (close to towns, villages, resorts) may be designated MPAs and highly overfished, while isolated reefs may not be designated as MPAs, but are lightly fished. If the authors have no data regarding MPA enforcement, then they need to make it very clear that they did not assess MPA effectiveness, and tone down their conclusions accordingly.

And as for factors influencing coral cover, I don’t know what to think of the HII metric. With the rapid development of resorts on the islands along the BBR during the study period, increases in cruise ships, live-aboards, etc., it’s not clear how the HII dataset truly represents anthropogenic impacts. I’m not arguing against the analysis, just that it’s easy to poke holes in this data set, and the authors may want to acknowledge its limitations.

On the topic of broad-scale stressors, the authors emphasize heat waves and climate change (L88), sometimes linking this with disease (L73), and often not (L353). In fact, there is no conclusive link between heat stress events and disease events, and the latter may be the result of anthropogenic introductions (e.g., ballast water), flare-ups of virulent disease variants, etc. This is important because these two things (climate, disease) are distinctly different, and this paper vaguely merges them together. Further, the data in this manuscript are now 5 years old, and in that time SCTLD has ravaged reefs across the Caribbean. It is likely that, even if there had been no decline in coral cover over the study period, it would have begun dropping precipitously from SCTLD in the past 2 years, and plunged below 10% without warming events playing any role. Interestingly, SCTLD stops spreading under warm conditions, suggesting it is not tied to warm water events. So, how does SCTLD play into this paper’s recommendations (L50-52)? What is the point of MPAs and reduced emissions when reef-building corals are functionally (reproductively) extinct on the vast majority of Caribbean reefs? Are studies such as these for the history books, or to guide us as Indo-Pacific reefs increasingly progress in the same way as those in the Caribbean?

It was surprising that the authors didn’t look more closely at macroalgal cover as a function of MPA status. Was seaweed cover higher inside or outside of MPAs (at any one time point, over several, etc)? This variable has been targeted in many other studies, both because seaweeds grow rapidly and overfishing removes herbivores. Looking at the red and blue dots in Fig 2, there doesn’t seem to be a consistent pattern for macroalgae across the time-series, but it would be interesting to know if there is an effect of MPAs (despite the lack of MPA validation).

I may have missed reference to this, but were the authors planning on providing their raw data somewhere? I didn’t see a link in their SI.

Reviewer #3: Overall, the paper is good, and I support publication. The comments below are to help the editor/authors consider various specific points. A few points are more confusing than others and need to be more clearly addressed.

Line 78. Please provide direct refs for “nutrient loading” as a secondary driver. I think this is controversial although I’ve not checked recent literature, and believe it needs a little more “play” in this introd.

Line 81-83. I don’t disagree that the crux is about relative importance (local vs regional/global). This is key and I’m glad your paper is focusing on this. But, it seems to me a that looking for various ‘direct effects’ is missing a big point, that there could be ultimate causes (climate, for ex) and then secondary causes related to the ultimate by INDIRECT effects. A conceptual model addressing this would be a good addition if links are supported enough to be causal, but this is really a relatively small point re: your paper.

Line 130-131. I know what you mean, but you might could say it more clearly or directly. You swam a 30 m long transect over a period of 5-7 min (depending on swimmer, and conditions I suppose….) and took photo…etc. Is less convoluted in description.

Lines 172-174. “We extracted HII values for the BBR (Fig. S1) and calculated the sum of the HII 173 scores of grid cells within a 50-km, 75-km, and 100-km buffer from the center-coordinates of 174 each study site (Table S1). We used HII scores within the 50-km buffer for the final analysis…” This bothers me some because the smallest scale was the one chosen, which presumably showed interesting results. But, the values are down to 1 km2 scale, so it would seem as if smaller scales should have been investigated as well. Large scales might unduly weight against the local factors that you argue against later. If this isn’t the case, you need to MAKE the case!

Lines 184-185. “Other studies have found that TSA Freq is a significant predictor 185 of coral-cover loss and coral-disease prevalence (62–64).” Ok, but is it a good predictor or a weak one? R2 or measures of effect size needed since I’ve (and other readers) won’t have read 62-64.

Line 193. “…logit-transformed percent covers of key benthic categories.” Which were what????? You go on to list predictors in some detail but not response. Is this coral cover? By Species? By Genera? By functional group? Etc, etc. more detail please, after all it’s your RESPONSE variables!

Lines 196-7. “…blme prior with a wishart distribution was imposed over 197 the covariance of the random effect and modeled coefficients”. Why Wishart? Were any others tried? Give me a reason to accept that this is a good analysis, with some ways to back this up and let me go check on my own if I wish.

Lines 197-8. “All predictor variables were 198 additive…” I assume because you CHOSE not to test for interactions? Is that correct? If so, say so, and defend why.

Lines 200-208. So…it looks like that Year is the factor relating to climate change. Is that right? But TSA freq is also in the model, are they not correlated and thus measuring some of the same variance in your response??? TSA freq would seem to increase with year, does it not? Need to discuss this to quite the drums….

Lines 263-264. “….ordination analysis, there were major compositional shifts in the dominant benthic 264 assemblages during 1997–2005 (left) and 2009–2016 (right) at every site (Fig. 6, Table 3), 265 supporting the results of our models.” Please explain more the L and R aspect of this and how it supports the results of your model. Also, I presume you mean that “support the results of your model” means that you posed certain models (additive only…but ok) a priori and that this is what is supported. It might be more clear to earlier say your models are posed as a priori hypotheses, and later discuss your results in light of the HYPOTHESES rather than relating directly to MODELS. A small point, actually.

Discussion: lines 308-318. I “get” what you’re saying but your analysis + explanation leaves me a little cold. First, you have temp (TSA freq) which you ‘bought in to ‘ as a global factor. But then it has little if any effect. But, time does….so basically you seem to transfer your time argument from TSA to time per se. If I’ve missed the mark here, then it means your explanations at several points need revising and expanding…because that’s the way it seems to me. I guess the point is: if you have TSA freq which is presumable increasing over time, why have time in the model too??? I suspect there’s a good reason, but I didn’t see it in the text.

Reviewer #4: This is a significant amount of reef data collected over a long time period. However, can such bold statements about the inutility of local regulations be based on 15 survey sites that are 30x2m wide? Coral reefs have such diverse environmental conditions and influencers of health. Here are my comments that the authors should address:

1. You cannot estimate local impacts with global datasets - there is a clear scale mismatch and you cannot infer local community status.

2. There was no investigation into the quality of the HII dataset which can underrepresent human impacts at the local scale, especially in less developed countries since many of the inputs are remote sensing-based and cannot be detected with coarse resolution data. Was there any vetting of the data with local experts to see if the changes detected in the HII approximated reality? The year-to-year changes need to be vetted with locals. HII is purely land-based and many of the sites are on islands that are not accurately detected with HII, are far from the coastline, thus limiting influence. HII is a land-based model and ocean sites also require a marine-based threat model to properly determine impacts.

3. Since there were many storms and hurricanes over the study time period, these can act as a catalyst for breaking up corals which can colonize elsewhere and stimulate growth. It is difficult to detect the changes in coral cover across the reef if you are going back to the same survey transects and only looking at the same 30x2m area.

4. There are many papers that demonstrate MPAs have a positive effect on coral cover. (Strain et al 2019; Strain et al 2019; Mellin et al 2016;Magdaong et al 2014; Selig and Bruno 2010;Rogers 2009). Did the authors look at management effectiveness for these parks and were there surveys carried out for the MPAs? Do they think that since an MPA is in place, it is actually preventing people from fishing? MPAs will not work if enforcement and management is not carried out.

5. While reducing emissions is the number one priority, the authors should acknowledge that saving coral reefs will require a number of local actions including better watershed and fisheries management, restoration of coastal habitats, and active restoration that focuses on restoring the natural recovery processes and genotypes that are the most resistant to disease and bleaching. You cannot dismiss the importance of local regulations (reserves, reduced fishing pressure, limiting development) which have been shown to have a positive effect on reefs.

6. PLOS authors have the option to publish the peer review history of their article (what does this mean?). If published, this will include your full peer review and any attached files.

Reviewer #1: No

Reviewer #2: No

Reviewer #3: No

Reviewer #4: No

---

## [Decision Letter · Decision Letter 1]

31 Aug 2021

Twenty years of change in benthic communities across the Belizean Barrier Reef

PONE-D-21-07319R1

Dear Dr. Bruno,

We’re pleased to inform you that your manuscript has been judged scientifically suitable for publication and will be formally accepted for publication once it meets all outstanding technical requirements. Please read the comments and see if any of my questions can be clarified addressed in the final version.

Kind regards,

Loren

Loren D. Coen, Ph.D.

Academic Editor

PLOS ONE

Additional Editor Comments (optional):

After receiving critical responses from two of the four past reviewers who had major reservations I recommend that the manuscript be accepted with minor revisions. A lot of the concerns by reviewers and myself have been addressed. Unfortunately, some cannot be without a major reanalysis of the captured images and other related parameters. A lot of the confusion and questions about the sampling scheme and subsequent analyses are now explained, however some the the critical info is in the responses but not in the revised text. I would have liked to see an expanded Intro/Discussion (though latter much better) albeit folks will have to glean a lot from past and other work. One tack would have been some of the prior work before the 20 years sampled here.

In the revised MS. in yellow means what??

A few relevant comments (by pg # as the author's responses are not numbered):

1) In the supporting letter the authors suggest that the reviewers and editor might want to move Table 1 to the Appendix. Rather, we all would like to see as much information as possible to be able to interpret your effort.

2) Given that there were extensive community images collected, I do wish that one of the coauthors could have been an algal person as this is in my mind something the dataset is lacking in overall rigor.

3) The title 'benthic communities' to me is still a bit misleading given the depth (or lack thereof) of the included categories and related species, with nearly all attention being directed to coral and gorgonian species resolution (as expected by the primary author's expertise and interests). As a benthic person that has worked with all of the components I would have liked to have seen more focus on non-coral benthic components (lines 125 on).

Yes it was. We added this detail to the methods: “At each site, we photographed or videotaped the belt transects at a standard distance of 25 cm above the benthos, using a bar projecting from the front of the camera housing to maintain distance from the bottom.” (L. 132–134). The depth of the analysis of benthic components is quite uneven would you not agree? Twenty years of change in benthic communities across the Belizean Barrier Reef perhaps could be changed to Twenty years of change in coral reefs along with their associated communities across the Belizean Barrier Reef?

4) Fleshy and corticated algae and Other inverts are barely identified (generic name or to sp. or Macroalgae, Tunicate, Bivalve, etc.) in S2 Table and the related analyses. With all the work in Belize by Littlers and Norris' I expect more. You state Lobophora and Dictyota primarily. How about percent cover from other papers from 1980s on? Lines 152 (categories) " 2) macroalgae, including calcareous, filamentous, corticate and/or leathery algae;..."

Really not a rigorous evaluation of these groups. Turfs see prior work there by Lewis or Littlers? What would Steneck say???

5) Pg. 6, state "We agree! We should have discussed changes in algal cover, over time, with protection, etc. We have added text to the Methods, Results, and Discussion and this new graphic:" I do not see this new graphic referred in either the revised MS or the revised Suppl. Materials?? My concern is not with the analysis itself but the depth of the data collected (categories and spp.) as outlined.

6) Same with Crustose algae, did you see if where live corals dominated early on?

7) Would have like to see more detailed general observations from folks that have worked so long in Belize. Are any data available to support that summer trends are indeed typical of patterns that one might observe year round in other seasons?

I understand the constraints you faced teaching and limited funding, however you need to convince the readers if possible that the documented trends are typical and consistent (not for long lived corals but other more ephemeral species or individuals).

8) Of course there are 1000s of papers seeing similar patterns again the focus here is a unique and long term 20 year dataset at a limited number of sites along the Belizean Barrier Reef. You wrote "Among us, we’ve written about these topics in many, many papers" for the uninitiated or readers new to subject want to make sure they understand the related information in these papers.

9) HII and its use here is now clearer to us and I hope also to readers unfamiliar with this index. Perhaps include a link ( perhaps https://sedac.ciesin.columbia.edu/data/set/wildareas-v2-human-influence-index-geographic)? I do think that it may not be as useful for cayes 30 miles offshore however.

10) I can understand your not seeing Sargassum on reefs, it floats obviously and is accumulating on caye shorelines everywhere.

11) Removal of mangroves and their filling in around the offshore cayes is depleting juvenile habitat at an alarming rate. SAV is also being lost often 100% around some of the cayes i visiting fairly recently. Extensive Sargassum is dying onshore and pumping vast amounts of nutrients into these offshore systems. With SAV lost the sediments are no long bound and I saw directly a soup of rotting rhizomes where extensive beds resided only a few years ago at these same locations.

12) Pg. 7, Regarding Diadema or other herbivores, would have loved to see some data here in cited in other papers of yours documenting their abundances. Stete "Herbivory has declined primarily because of the loss of the black sea urchin Diadema antillarum due to a regional disease outbreak (31) and severe reductions of populations of herbivorous fishes due to fishing” (32–37)". I agree to some extent but Diadema declined prior to mid 1980s and yet corals were in relatively good shape then (pers. obs). They have come back but never to the extent they once were then.

13) Authors state "Long data sets such as this are comparatively rare. While the original intent was not to publish a long term dataset when the project started, the data were collected for various reasons and it makes sense to present it as a long look back." If this is where you are synthesizing a lot of the information or related data from all of this work the manuscript should reflect the data's quality, as well as quantity no??

14) Pg. 5, If you believe the data that most no-take MPAs in Belize or across the Caribbean or at a wider lens have minimal or no effect on either fish biomass and other metrics does that negate their value if they could work with proper regulation? Given that seems like a red herring to first include and then wave away as potential tool for protecting fisheries and upstream effects, no?

15) Table S1 (Pg. 3) are inadequate to Good the extent of the compliance categories? This is not clear in the Table provided with related text in the responses.

16) The methods are much improved (pg. 7). State "The transects generally began on or near the shoulders of the spurs at 15–18 m depth, shore ward of the drop-off that characterizes most of the reefs, and ran upward toward the reef crest." Want to make sure that the spurs are deep spur zone and not shallower spur and groove zone? Most drop-offs began at what depth? Clearly state all your sampling was within the forereef zone??

17) Pg 12 glad that you see why confusing and have clarified this in the text (lines 311-318).

18) Pg. 13 state and provide a figure 3. Why not cite see (Bruno et al. 2001) Fig 3 in text?

19) Pg. 13, Item 2, We are unaware of information supporting this argument. The most important component of HII (human population density) is not measured remotely; it is based on national on-the-ground census data (via the Statistical Institute of Belize: https://sib.org.bz/publications/census-reports/). I think that it is reasonable despite the lack of info that what is of concern might indeed be relevant? See comment #11 above and your comments on pg. 16, "All that said, we fully recognize the limitations of this measure of local human impacts. It does not, for example, include dredging near San Pedro (and elsewhere) to expand islands for development. It does not include potential impacts of cruise ships not anchored at ports. We wish it was better." as an example.

20) I agree that folks on this paper are well versed in Belize but many with the exception of Aronson were not there in early 1980s and before. Perhaps some of the observations of others (see Carrie Bow crowd then) might have seen some of the forest for the trees (focus of research of the authors here)?

21) Not just murky water (although even 5 years or less many previously visited spots have declined precipitously. Most have not seen the backreef around SWC area including CBC the way it used to be. Porites heads gone, Twin Cayes nothing like it used to be before all the boat disturbances, etc.

22) Item 4, Pg. 16-17 on. Why not include more of your comments and cite Table 1 in Discussion?? There were really no MPAs in Belize in 1980s and before. Lobster and conch fishing were poorly regulated. Most of the former were trapped prior to the opening of the season. Many fish species were rarely seen even in the latter part of 1980s despite being unfished. State "Note that most papers cited by the reviewer demonstrate that MPA have a weak positive effect on total coral cover under very specific conditions" remember that your title reflects changes in overall benthic community not just the coral component no?

23) Section revised (pg. 19) goes a long way to making the argument less dismissive.

24) Rather than suggesting readers see a MS thesis state the relevant conclusions, etc.

25) Pg. 20-21, State "No. This study was originated in 1997 by coauthor McField as a single, comprehensive assessment of the state of Belize’s reefs to focus on 6 sites. The project then was expanded the following year as a long-term monitoring study with more sites (See Table S1). Due to funding limitations and other logistical constraints (many co-authors needing to teach during the academic year), we were not able to access every site every year (or every sampling trip). From the pool of sites, those selected on a given trip were somewhat haphazard, often depending on weather and safety concerns, although we always sampled close to an equal number of protected and unprotected sites each year." This was never clear in the original MS and I hope the readers understand this now (vs. clarifying in your responses vs. the revised text)!

26) Pg. 21, "Ln 129 Was the GOPRO camera fitted with the 25 cm rod? GOPROs have fisheye type lenses which change the analysis area.

Yes it was. We added this detail to the methods: “At each site, we photographed or videotaped the belt transects at a standard distance of 25 cm above the benthos, using a bar projecting from the front of the camera housing to maintain distance from the bottom.” (L. 132–134)."

We wondered whether the field of view of the various cameras and GoPros were similar? Ln 129 Was the GOPRO camera fitted with the 25 cm rod? GOPROs have fisheye type lenses which change the analysis area. Distance from substrate vs. the actual field of view with different lenses? Very different thing. Scale of the data and area selected when you did the CPC would be different no? Never provided that I can find. Is that available?

27) Were any changes made in the Suppl. document see no obvious marked up version 2 with tracked changes??

28) Finally page 22-25 of your responses are these new citations or just in support of the responses? A cursory comparison of the two files seemed to show in there already?

Reviewers' comments:

Reviewer's Responses to Questions

**Comments to the Author**

1. If the authors have adequately addressed your comments raised in a previous round of review and you feel that this manuscript is now acceptable for publication, you may indicate that here to bypass the “Comments to the Author” section, enter your conflict of interest statement in the “Confidential to Editor” section, and submit your "Accept" recommendation.

Reviewer #1: All comments have been addressed

Reviewer #2: All comments have been addressed

Reviewer #4: All comments have been addressed

2. Is the manuscript technically sound, and do the data support the conclusions?

Reviewer #1: Yes

Reviewer #2: Yes

Reviewer #4: Yes

3. Has the statistical analysis been performed appropriately and rigorously? 

Reviewer #1: Yes

Reviewer #2: Yes

Reviewer #4: Yes

4. Have the authors made all data underlying the findings in their manuscript fully available?

Reviewer #1: Yes

Reviewer #2: Yes

Reviewer #4: Yes

5. Is the manuscript presented in an intelligible fashion and written in standard English?

Reviewer #1: Yes

Reviewer #2: Yes

Reviewer #4: Yes

6. Review Comments to the Author

Reviewer #1: (No Response)

Reviewer #2: This is Reviewer 2 from the first version.

The authors have done a good job of addressing my comments from the first version, particularly:

(1) I like the broadened data and discussion regarding MPAs and relative effectiveness,

(2) Better explanation of HII and how it was used,

(3) references to the ravages of SCTLD, and how much of this debate may be moot,

(4) addition of info on macroalgae, which will be around for some time to come.

Overall, I think the ms. is more circumspect, and is a better vehicle for presenting an important set of time-series data.

Reviewer #4: My concern about the limited number of sites has been adequately addressed since the authors document hundreds of other sites showing similar patterns. I agree that global scale datasets have indeed been used to infer local impacts as the author’s suggest, however one of my main concerns was the lack of vetting and validating the data with local experts. The fact that the HII has been created using census data alleviates this concern. The added discussion in the text helps to further explain how HII is used and some of the caveats about their use of the metric. The authors provide a recognition of the limitations when assessing local impacts including dredging, cruise ship impacts, and sediment loads which is all difficult to obtain. That said, I agree that HII provides adequate surrogate estimates for human activity and potential stressors. The addition of random site analysis was directly in reference to detecting the benefit of increased generalization as the authors allude to. I agree that a discussion on or testing the effectiveness of MPAs is beyond the scope of this paper although the authors have added text describing some of the limitations of Belize’s MPAs. Clearly, more research is needed to better understand the net positive effect of no-take MPAs and relate the findings to enforcement levels. Revisions to the discussion on local actions and regulations further strengthen the paper and provide insight into their importance.

7. PLOS authors have the option to publish the peer review history of their article (what does this mean?). If published, this will include your full peer review and any attached files.

Reviewer #1: No

Reviewer #2: **Yes: **Joseph R Pawlik

Reviewer #4: No

---

## [Editor Report · Decision Letter 2]

29 Nov 2021

Twenty years of change in benthic communities across the Belizean Barrier Reef

PONE-D-21-07319R2

Dear Dr. Bruno,

We’re pleased to inform you that your manuscript has been judged scientifically suitable for publication after several revisions, and will be formally accepted for publication once it meets all outstanding technical requirements. Thank you for your patience and willingness to address most of the concerns past and present.  I think it is much improved.

Within one week, you’ll receive an e-mail detailing the required amendments (none from me).  When these have been addressed, you’ll receive a formal acceptance letter and your manuscript will be scheduled for publication.

Kind regards,

Loren D. Coen, Ph.D.

Academic Editor

PLOS ONE

Additional Editor Comments (optional):

The manuscript is acceptable in its final form.

Don't want to get picky just know several of the team and others have looked at their backgrounds and pubs (response to #2, How does the editor know one of the coauthors isn’t an algal person? How are they assessing the taxonomic expertise of the research team? We identified algal to genus when possible given the limitations on identifying macroalgae from still images extracted from videos). The Ms and Table S2 are at the barest level of identification and enumeration (appears no vouchers taken to assess video IDs of algae otherwise not all to Genus or higher no)?? Also if you disagreed assume you would specify person(s) no? Did not see any vouchers specimens placed in museum collections in text either. I think that you could have identified and enumerated Sargassum spp., Dictyopteris spp., Stypopodium spp., Turbinaria spp., Halimeda spp., Zonaria sp., Dictyota spp., Padina spp. all spp. safe bets as over the entire dataset not one image no?? I would rather see Dictyotales spp. as you cannot ID Dictyopteris vs. Dictyota by video. If you put Lobophora sp. how do you know its not spp. or L. variegata?? How valuable is listing 'macroalgae' or 'fleshy macroalgae' under macroalgae Table S2? That seems rather weak in terms of phycology? I guess I feel the way you might if I put brain coral, or Porites sp., or fire coral, etc. See my point?

Responses #3 & 4), "It is not possible to identify most benthic reef organisms (excluding corals) to species via still images or video or in situ (e.g., for most sponges, gorgonians, and seaweeds samples must be collected for examination under a microscope for reliable species-level identification). If your video is of decent resolution then an algal person should be able to ID to finer scale. Vouchers for later cross check. This is the scale of taxonomic resolution for most ecological monitoring in coral reefs." I agree that is why I have stressed that the paper deals primarily with corals and not the benthic community in general as the title suggests. Given I've spent many 100s of hours sampling macroalgal community along Belizean reef on Carrie Bow I don't agree that the macroalgal community is characterized. Functional groups fine for some things, but if concerned about smothering live corals or lack of grazers perhaps not best in my opinion. Let's agree to disagree.

I assume in item (#4) you are referring to Steneck, R.S. et al. 2019. Managing recovery resilience against climate-induced coral bleaching and hurricanes: A 15-year case study from Bonaire, Dutch Caribbean. Frontiers in Marine Sciences? That paper was a broad brush overview with focus on corals and grazers (see #2 also above). Still do not know the area of image assessed?? You omitted the latter group here (cf. your examples of Figs). He's after all a coralline guy, what I was suggesting is a person like Norris or Hanisak, or Fredericq? AGRRA protocol I would assume there minimum not all one might do?

Responses #5 & 6, Improved from original (Figs. 2 & 6). Even back to mid 1980s coral cover vastly different. Have images of quadrats along many sites near CBC with detailed ids of macroalgae.

Response #7, Agree that it would require more than swimming transects each time. Never asked for seasonal changes in algal communities. Rather suggested that work done on those groups in Belize could easily be cited and summarized/cf. given what you have as caveat and potential missing component (Fig. 1 McClanahan et al. showing me vs. readers of value). Note that McClanahan et al. have extensive algal IDs in Appendix 1! Assume why paper not submitted to Ecol. Monogr.?

Response #8, Many readers interested in your conclusions not familiar with Belize or even reefs may need that detail despite them not having 20 yr datasets.

Response #9, D The link to the HII data is already included in the references (# 57). The HII data also cover the cayes that are 30 miles offshore (see Fig. 1A & 1B). Did you perhaps mean on line 717, #59?

Response #10, All Sargassum does not float, why papers include in their lists (Targett et al. Oecologia, McClanahan et al., Littlers et al., Lapointe, Norris, etc.). Many start or remain attached.

Response #11, observation is iron clad, Ken Heck and I were there in 201 and then 3 years later no SAV (Thalassia) along shorelines mostly gone throughout a lot of coast of Ambergris.

Response #12, I good old days one saw them even in daytime on reef and backreefs in Belize in early 1980s.

Response #13, understand.

Response #14, Just want to be careful that folks don't misinterpret the short conclusion in abstract without reading further your takeaway message.

Response #15, clearer but want to provide earlier no than Shantz AA, Ladd MC, Burkepile DE. Overfishing and the ecological impacts of extirpating large parrotfish from Caribbean coral reefs. Ecol Monogr. I think? As late citing a reference?? State ) corresponding to each year that an Eco-Audit was performed.” (Lines 818-824)??

Response 16, This information is already in the Methods section a diagram might be nice but let's hope that folks know crest and drop off terms reading this. We were not all confused with that its a lot of the other stuff that was vague. Don't like to guess with Methods. Authors familiar with all the nuances of their sampling but readers are not.

Response 17, OK

Response 18, Good

Response 19, I think some of us concerned that the population #s do not capture the strain being put on Cayes along the barrier reef when using overall population #s. Leave it it will have to do.

Response 20, Ok, most of citations were from work much later that is all. Leave it as is.

Response 21, OK

Response 22, does not mention larger inverts understandably.

Response 23, Ok

Response 24, Good

Response 25, OK

Response 26, You give reader distance but not area imaged why not??? What is the range of field of imagery?? If we put down a frame (quadrat with 10 cm divisions) what would we se in the image you quantified)?

Response 27, OK

Response 28, OK
---

## [Editor Report · Acceptance letter]

7 Jan 2022

PONE-D-21-07319R2 

Twenty years of change in benthic communities across the Belizean Barrier Reef 

Dear Dr. Bruno:

I'm pleased to inform you that your manuscript has been deemed suitable for publication in PLOS ONE. Congratulations! Your manuscript is now with our production department. 

Kind regards, 

on behalf of

Dr Loren D. Coen 

Academic Editor

PLOS ONE